# Pathway selectivity in Frizzleds is achieved by conserved micro-switches defining pathway-determining, active conformations

Lukas Grätz [1,6], Maria Kowalski-Jahn [1,6], Magdalena M. Scharf [1], Pawel Kozielewicz [1], Michael Jahn [2,3], Julien Bous [1], Nevin A. Lambert [4], David E. Gloriam [5] & Gunnar Schulte [1]✉

The class Frizzled of G protein-coupled receptors (GPCRs), consisting of ten Frizzled (FZD_{1-10}) paralogs and Smoothened, remains one of the most enigmatic GPCR families. This class mediates signaling predominantly through Disheveled (DVL) or heterotrimeric G proteins. However, the mechanisms underlying pathway selection are elusive. Here we employ a structure-driven mutagenesis approach in combination with an extensive panel of functional signaling readouts to investigate the importance of conserved state-stabilizing residues in FZD_5 for signal specification. Similar data were obtained for FZD_4 and FZD_{10} suggesting that our findings can be extrapolated to other members of the FZD family. Comparative molecular dynamics simulations of wild type and selected FZD_5 mutants further support the concept that distinct conformational changes in FZDs specify the signal outcome. In conclusion, we find that FZD_5 and FZDs in general prefer coupling to DVL rather than heterotrimeric G proteins and that distinct active state micro-switches in the receptor are essential for pathway selection arguing for conformational changes in the receptor protein defining transducer selectivity.

Constitutive or ligand-induced activation of G protein-coupled receptors (GPCRs) is accompanied by an overall change in receptor conformation resulting in hallmark rearrangements. This includes the outward movement (classes A and F) or unwinding (class B1) of the cytosolic region of transmembrane domain (TM) 6 opening the receptor pocket, which allows accommodating transducer proteins, such as heterotrimeric G proteins[1,2]. The activation of the major classes of GPCRs engages additional helix macro-switches, including movements of the cytosolic region of TM5 and rotations of TM3, and residue micro-switches, which are unique for each class, except for classes A and B1 sharing four such switches[2].

The class F of GPCRs consists of ten Frizzleds (FZD_{1-10}) and Smoothened (SMO), which employ diverse transduction mechanisms and have distinct features of receptor activation[3-5]. In humans, FZDs are targeted by 19 secreted lipoglycoproteins from the Wingless/Int-1 family (WNTs). The downstream signaling cascades are tightly regulated and vital during embryonic development as well as in adults, e.g. for tissue homeostasis and stem cell maintenance[5]. The predominant dogma of WNT-induced and FZD-mediated signal initiation is based on a signalosome model that depends on receptor heterodimerization in response to agonist stimulation, rather than ligand-induced conformational dynamics in the WNT-sensing receptors, the FZDs[6-9].

[1]Karolinska Institutet, Dept. Physiology & Pharmacology, Sec. Receptor Biology & Signaling, Biomedicum, S-17165 Stockholm, Sweden. [2]School of Engineering Sciences in Chemistry, Biotechnology and Health, Science for Life Laboratory, KTH – Royal Institute of Technology, S-17121 Solna, Sweden. [3]Max Planck Unit for the Science of Pathogens, Bioinformatics platform, Charitéplatz 1, D-10117 Berlin, Germany. [4]Department of Pharmacology and Toxicology, Medical College of Georgia, Augusta University, Augusta, Georgia, USA. [5]Department of Drug Design and Pharmacology, University of Copenhagen, Copenhagen, Denmark. [6]These author contributed equally: Lukas Grätz, Maria Kowalski-Jahn. ✉e-mail: gunnar.schulte@ki.se

However, focusing on understanding the receptor activation mechanisms in detail, we have previously identified dynamic conformational rearrangements in FZDs, which reach from receptor dimer dissociation[10], over conformational rearrangements within the receptor TM bundle[11–13], rearrangements in the FZD-transducer interface[14,15] to dynamics of the N-terminal cysteine-rich domain relative to the receptor core[16]. Importantly, a polar interaction between the conserved basic residue R/K$^{6 \times 32}$ and the backbone of W$^{7 \times 55}$ in FZDs serves as a common molecular switch with opposite impact on the binding of G proteins and the phosphoprotein Disheveled (DVL)[17]. Thus, despite the existing signalosome model, conformational dynamics of FZDs that are potentially independent of receptor oligomerization emerge as a crucial component in WNT/FZD-dependent signal initiation and pathway definition.

The three isoforms of mammalian DVL proteins (DVL1, 2, 3) function as protein scaffolds at the crossroads of FZD-induced signaling and mediate both β-catenin-dependent and -independent pathways[18,19]. DVL interacts with FZDs mainly through the DEP (Disheveled, Egl-10, and Pleckstrin) domain, which is one of three conserved, structured subdomains of DVL proteins[18]. The isolated DEP domain interacts strongly with overexpressed FZDs in a constitutive and phosphatidylinositol(4,5)bisphosphate (PIP$_2$)-dependent manner, which can, for example, be assessed using microscopy- or more recently also bioluminescence resonance energy transfer (BRET)-based assays[9,12,14,17,20,21]. While WNT/β-catenin signaling appears independent of heterotrimeric G proteins[22], FZD-G protein coupling has emerged as a physiologically relevant and mechanistically better understood concept[13,15,17,23–27]. In agreement with the ternary complex model, selective active conformations of GPCRs in general, potentially also of FZDs, define the signaling outcome by allowing the accommodation of distinct intracellular transducer proteins, such as heterotrimeric G proteins, arrestins, GPCR kinases and possibly DVL[3,14,17,23,28,29].

Here, we test the hypothesis whether distinct receptor conformations also determine downstream signaling for the FZD family of receptors using FZD$_5$ as a representative. FZD$_5$ has served as model receptor to assess FZD-mediated WNT/β-catenin signaling[17,30], FZD-DVL interaction[14,20,30,31], FZD-G protein coupling[13,17], and conformational dynamics[11,13,14,16] and appeared therefore as a suitable target to focus on receptor-intrinsic mechanisms responsible for pathway selection. We employed extensive mutagenesis of potential state-stabilizing residues or 'micro-switches', which were selected based on a comparison of active and inactive class F GPCR structures. The receptor mutants were tested with a comprehensive palette of signaling readouts to understand how FZD$_5$ particularly and FZDs in general achieve transducer selectivity towards DVL over heterotrimeric G proteins. The unique combinatorial assessment of DVL- and G protein-focused readouts allowed an unbiased cluster analysis identifying amino acids and regions in the receptor protein that are important for pathway selection. Cell-based experiments were complemented by molecular dynamics (MD) simulations of selected FZD$_5$ micro-switch mutants, which corroborated the concept of receptor dynamics. As the mutated residues are highly conserved among the ten FZDs, our results might be extrapolated to the whole receptor family with class-wide implications. To further support class-wide conclusions, we complemented our experimental data with two additional representatives of the FZD family, FZD$_4$ and FZD$_{10}$, which provided similar insights. Thus, our findings suggest that downstream signaling of FZDs is indeed specified by different receptor conformations, which are stabilized by distinct residues and intramolecular interactions. This information will be valuable for a better understanding of FZD molecular pharmacology and subsequently for future drug discovery efforts aiming to design pathway-selective FZD-targeting drugs.

## Results

### Predicted state-stabilizing residues for mutagenesis of FZD$_5$

To predict state-stabilizing residues for class F GPCRs, we took advantage of a recent online platform for GPCR structure comparison[32] designed to uncover activation determinants across all major classes of human GPCRs[2] (detailed in Methods). We used a threshold value of 80% for the two complementary tools, 'Structure comparison tool' (cut-off value based on sequence conservation) and 'State-affecting mutation design tool' (cut-off value based on the percentage difference of net contact frequencies between the active and inactive structures), generating 25 and 16 predicted state-determinant residues, respectively (Fig. 1a). Out of those, four residues were alanines in our prototypic receptor FZD$_5$ and were therefore excluded from the planned alanine screen. Additionally, five of the residues were identical for both prediction tools, finally resulting in 32 distinct predicted putative state-stabilizing residues (Supplementary Data 1). These span all seven transmembrane helices (TM1-7), the second extracellular loop (ECL2) and helix 8 (H8; Fig. 1b). The high degree of conservation among these predicted state-stabilizing residues in class F receptors and especially in FZDs is visualized in a sequence alignment of all human FZDs and SMO (Supplementary Fig. S1).

The 32 alanine mutations were introduced into a mammalian expression vector encoding human FZD$_5$ carrying an N-terminal HA tag and a C-terminal 1D4 tag. For a first validation, we quantified cell surface expression by employing a whole-cell enzyme-linked immunosorbent assay (ELISA) detecting the extracellular, N-terminal HA epitope of the wild-type FZD or mutated receptors. For six of the 32 predicted micro-switch mutants, expression levels fell below the detection limit of the method (Fig. 1c, red labeling; Supplementary Table S2). Some of these were, however, still able to induce robust β-catenin-dependent signaling (Supplementary Fig. S2) emphasizing that even low surface expression levels are sufficient to mediate responses in readouts with high signal amplification. The other 26 mutants showed a distinct surface expression and could be further characterized regarding their impact on FZD$_5$ function.

### Effects of FZD$_5$ micro-switch mutants on the electrophoretic mobility shift of DVL

The scaffold protein DVL is an essential component in WNT signaling relaying both β-catenin-dependent and –independent pathways[18]. FZD$_5$ overexpression results in a characteristic, phosphorylation-dependent electrophoretic mobility shift of DVL even in the absence of WNTs[31,33]. Here, we investigated the effect of the FZD$_5$ micro-switch mutants on the constitutive, receptor overexpression-induced electrophoretic mobility shift of DVL2, which is readily detectable at endogenous expression levels by an isoform-selective polyclonal anti-DVL2 antibody using immunoblotting (Fig. 2a). The ratio of the densitometry signals of shifted (phosphorylated and hyper-phosphorylated) over basal DVL2 was used as a readout, and it revealed differential effects of the receptor mutants (Fig. 2b). The mutants T$^{1 \times 50}$A, T$^{1 \times 53}$A, and Y$^{2 \times 39}$A in TMs 1 and 2, F$^{4 \times 45}$A in TM4, R$^{6 \times 32}$A in TM6 and W$^{7 \times 55}$A, and W$^{8 \times 54}$A in TM7 and H8 shifted DVL2 significantly less than wild-type FZD$_5$. In contrast, the mutants L$^{5 \times 62}$A and G$^{5 \times 65}$A in TM5, G$^{6 \times 34}$A and L$^{6 \times 52}$A in TM6, and M$^{7 \times 44}$A in TM7 induced significantly stronger phosphorylation (hyper-phosphorylation) of DVL2 compared to wild-type FZD$_5$.

### Mutation of state-stabilizing residues affects WNT-3A-induced β-catenin signaling

One of the DVL-mediated WNT signaling outcomes is manifested in the stabilization of the transcriptional regulator β-catenin in the cytosol, which is subsequently translocated to the nucleus to regulate TCF/LEF (T cell factor/lymphoid enhancer factor)-dependent gene transcription. Activation of this WNT/β-catenin signaling branch, e.g. by

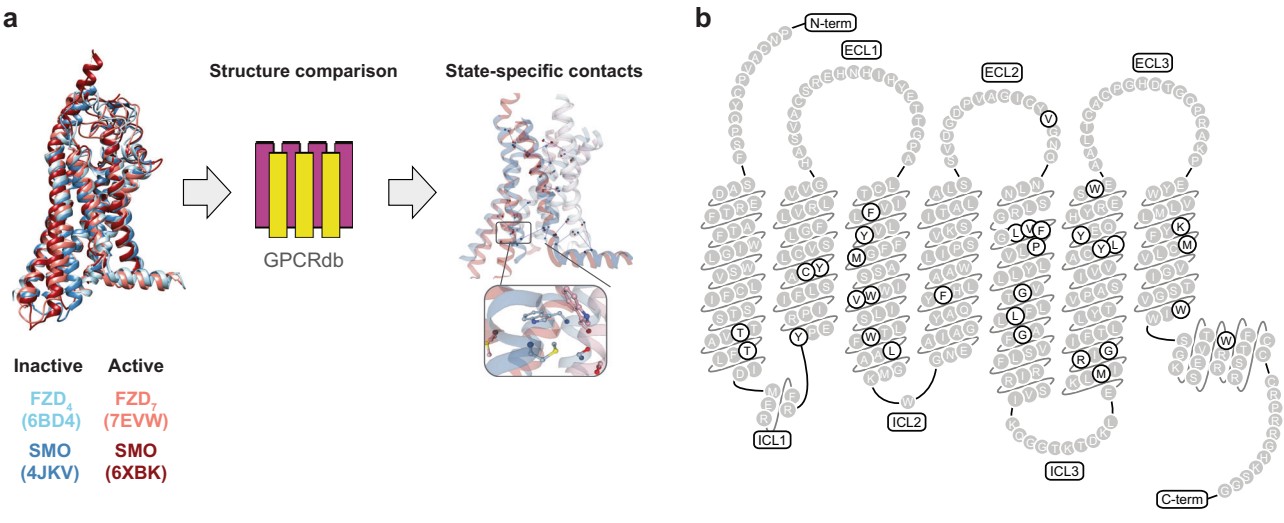

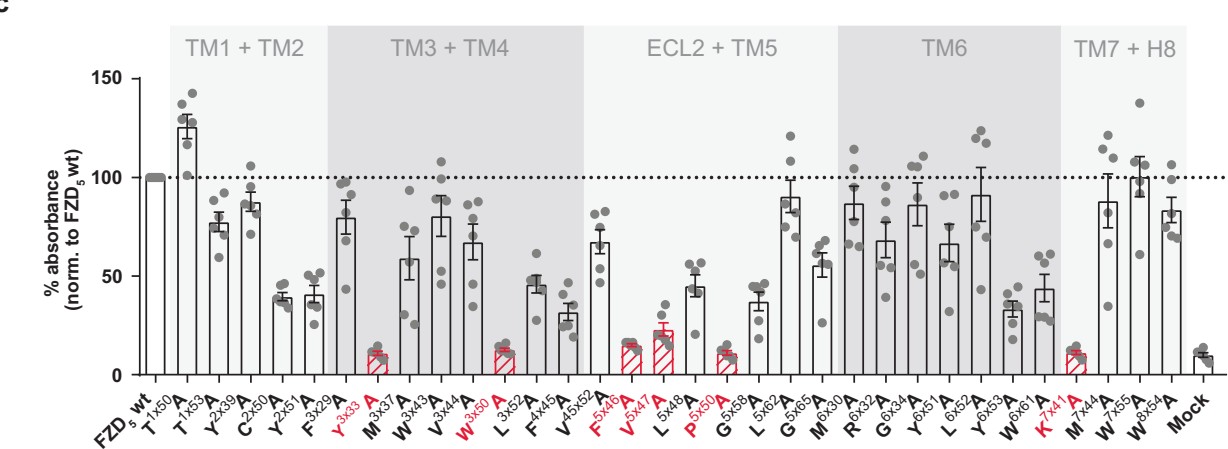

**Fig. 1 | Schematic overview of the investigated micro-switch mutants in FZD₅ and validation of mutants. a** Design of state-stabilizing mutations. Inactive and active state structures were compared using two GPCRdb tools[32] (see Methods) to identify state-specific residue-residue contacts formed by state determinant residues (Supplementary Data 1). **b** Snake plot of human FZD₅ with mutated amino acid residues highlighted in gray. The N and C termini of FZD₅ were omitted for clarity. **c** Validation of cell surface expression in HEK293A cells transiently transfected with the different FZD₅ micro-switch mutants, wild-type FZD₅, or pcDNA3.1 (Mock)

quantified by whole-cell ELISA using an antibody against the N-terminal HA tag. Data show mean ± SEM of six independent experiments performed in triplicate and mean values were normalized to wild-type FZD₅ surface expression. Results were analyzed with one-way ANOVA (matched, with Geisser-Greenhouse correction) and uncorrected Fisher's LSD post-hoc test. Note that only mutants labeled in red are not significantly different from pcDNA3.1 control. Corresponding significance levels are shown in Supplementary Table S2. C-term C terminus, ECL extracellular loop, ICL intracellular loop, N-term N terminus, TM transmembrane domain.

WNT-3A, is typically assessed with a luciferase reporter gene assay called TOPFlash (Fig. 3a)[34]. As expected, recombinant WNT-3A elicited a strong FZD₅-dependent TOPFlash response in wild-type FZD₅-transfected ΔFZD₁₋₁₀ HEK293T cells[35], which served as a reference point for the comparison with the different FZD₅ micro-switch mutants. While none of the mutants enhanced the WNT-3A-induced TOPFlash response, many mutants showed an impaired response (Fig. 3b). Most strikingly, the $R^{6×32}A$ molecular switch mutant completely abrogated the ability of WNT-3A to elicit a β-catenin-dependent transcriptional response similar to what was observed before[7,17], whereas the corresponding mutation of the counterpart in TM7, $W^{7×55}A$, led to a very low, but not completely blocked TOPFlash response. Along the lines of the results from the DVL shift assay, we also observed a strongly reduced signal for mutants $T^{1×50}A$ and $T^{1×53}A$ in TM1, $Y^{2×39}A$ in TM2 and $W^{8×54}A$ in H8. For the last two, this matches with previous literature reports, where both $Y^{2×39}$ in the conserved YPERPI motif as well as $W^{8×54}$, located in the KTxxxW motif, have been shown to be important for the interaction of FZDs with DVL and for induction of β-catenin-dependent signaling[36,37]. Interestingly, we also found a lower TOPFlash response

for many of the mutants with a strong DVL shift ($L^{5×62}A$, $G^{6×34}A$, $L^{6×52}A$, $M^{7×44}A$).

**Constitutive recruitment of the isolated DEP domain is affected by micro-switch mutations**

The DEP domain of DVL is essential for its transducer function and the interaction with FZDs[14,20,30,38]. Association of DVL or the isolated DEP domain with FZDs occurs constitutively upon overexpression of FZDs, i.e., in the absence of WNTs[9,14,20,21]. Following up on the results from the detection of the electrophoretic mobility shift of DVL, we were interested in determining whether the panel of FZD₅ mutations also affects the basal recruitment of the isolated DEP domain of human DVL2 to the receptor. Therefore, we made use of a direct BRET assay, tagging the C terminus of FZD₅ (wild type or mutants) with the bright Nano-luciferase (Nluc) as the BRET donor and the isolated DEP domain with the BRET acceptor mVenus (Fig. 4a)[14].

After confirming that all FZD₅-Nluc micro-switch mutants were expressed at the cell surface by whole-cell ELISA (see Supplementary Fig. S3 and Supplementary Table S2), we performed titration

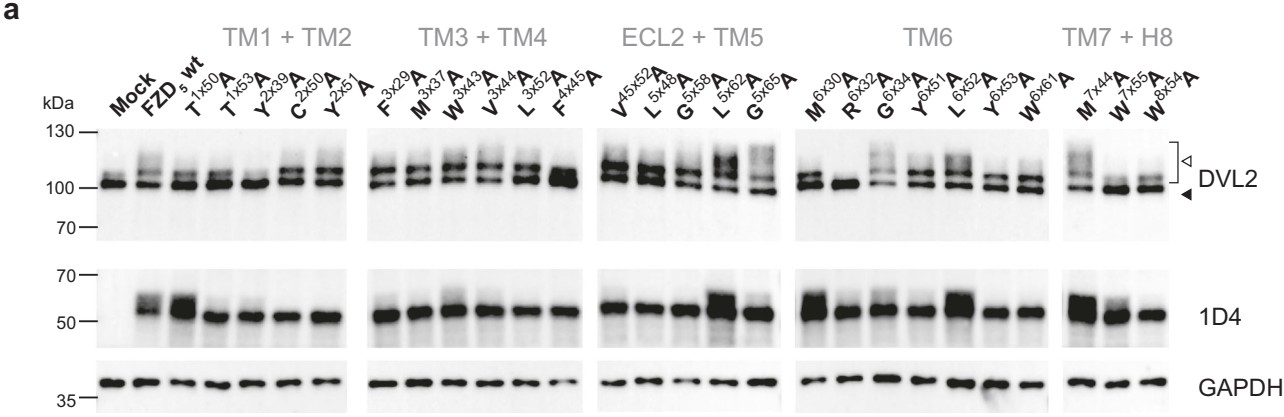

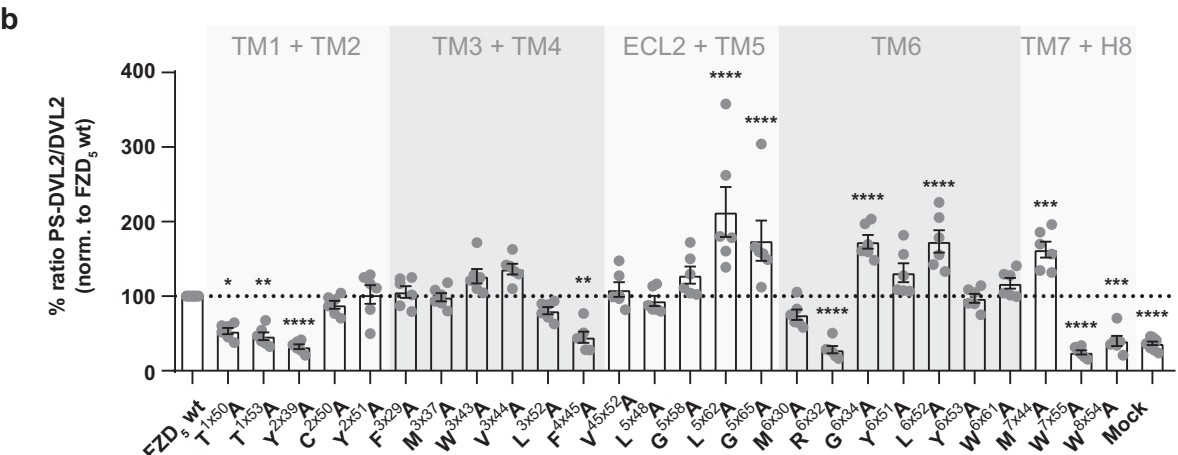

**Fig. 2 | Effects of FZD₅ micro-switch mutants on the electrophoretic mobility shift of DVL2. a** Immunoblots of HEK293A cells transiently transfected with the different FZD₅ micro-switch mutants, wild-type FZD₅, and pcDNA3.1 (Mock). Cell lysates were analyzed for the FZD₅-induced electrophoretic mobility shift of DVL2 from basal (bottom band; filled triangle) to shifted (bracket, open triangle) using an anti-DVL2 antibody to detect endogenously expressed DVL2. Expression of full-length FZD₅ constructs was detected using an anti−1D4 antibody. Anti-GAPDH served as a loading control. Note that the immunoblots are cropped. Wild-type FZD₅ and Mock were included on each individual blot as a control. Uncropped blots are shown in Source Data File. **b** Quantification of basal and shifted DVL2 bands by densitometry. Data show mean ± SEM of six independent experiments for the ratio of phosphorylated and shifted (PS)/basal DVL2. Mean values were normalized to wild-type FZD₅ on the respective immunoblot. Statistical differences between the micro-switch mutants and wild-type FZD₅ were assessed by one-way ANOVA followed by Dunnett's post-hoc analysis. Significance levels are given as follows: *$p < 0.05$, **$p < 0.01$, ***$p < 0.001$, and ****$p < 0.0001$. TM transmembrane domain, ECL extracellular loop, H8 helix 8.

experiments with all mutants, keeping the amount of FZD₅-Nluc plasmid DNA constant, whilst increasing the amount of the plasmid encoding the BRET acceptor DEP-Venus and monitoring their interaction by measuring BRET (Fig. 4b–f). While all FZD₅ micro-switch mutants showed specific interaction with the isolated DEP domain, which is reflected in a hyperbolic curve shape, they differed in their affinities (BRET₅₀; (Fig. 4g)) and/or maximal BRET values (BRET_max; (Fig. 4h)). As neither of these parameters depended on receptor expression levels (see Supplementary Fig. S4), they could be interpreted as mutant-intrinsic properties (Supplementary Table S3). By definition, a high BRET₅₀ value is equal to a low FZD₅-DEP affinity, whereas a higher BRET_max stands for more efficient energy transfer, reflecting either an interaction with more DEP molecules or a difference in conformation leading to a more favorable positioning of the BRET acceptor (DEP-Venus) to the BRET donor (FZD₅-Nluc).

While the tested mutants showed a wide array of values for both parameters, none of the investigated mutants displayed a significantly lower BRET₅₀ value, i.e., a higher DEP affinity (Fig. 4g), or significantly higher BRET_max value than wild-type FZD₅ (Fig. 4h). However, we could

identify some mutants, e.g., the mutants inducing a strong DVL shift (L⁵ˣ⁶²A, G⁵ˣ⁶⁵A, G⁶ˣ³⁴A, L⁶ˣ⁵²A, M⁷ˣ⁴⁴A), for which both BRET₅₀ and BRET_max values were not significantly different from wild-type FZD₅. In line with the results from DVL shift and TOPFlash assays, mutants T¹ˣ⁵⁰A, T¹ˣ⁵³A and Y²ˣ³⁹A also exhibited a lower affinity towards the DEP domain, i.e., higher BRET₅₀ values. The obtained BRET_max values were lower (T¹ˣ⁵³A, Y²ˣ³⁹A) or not altered at all (T¹ˣ⁵⁰A). Notably, mutating W⁸ˣ⁵⁴ in the KTxxxW motif, which is important for mediating FZD-DVL interaction[37], resulted in a slightly lower BRET_max value but without changing the affinity between FZD₅-Nluc and DEP-Venus. As expected, the molecular switch mutant R⁶ˣ³²A and to a smaller degree also W⁷ˣ⁵⁵A, the corresponding mutant in the molecular switch in TM7, both showed a very weak affinity to DEP and a lower BRET_max, even though the R⁶ˣ³²A mutation did not abrogate the interaction as it did for full-length DVL2 in a bystander BRET setup[17]. We also found the same pattern, i.e., higher BRET₅₀ and lower BRET_max, for mutants M³ˣ³⁷A, F⁴ˣ⁴⁵A and Y⁶ˣ⁵³A.

Collectively, by using our direct BRET assay, we were able to differentiate the FZD₅ micro-switch mutants in terms of their affinity to

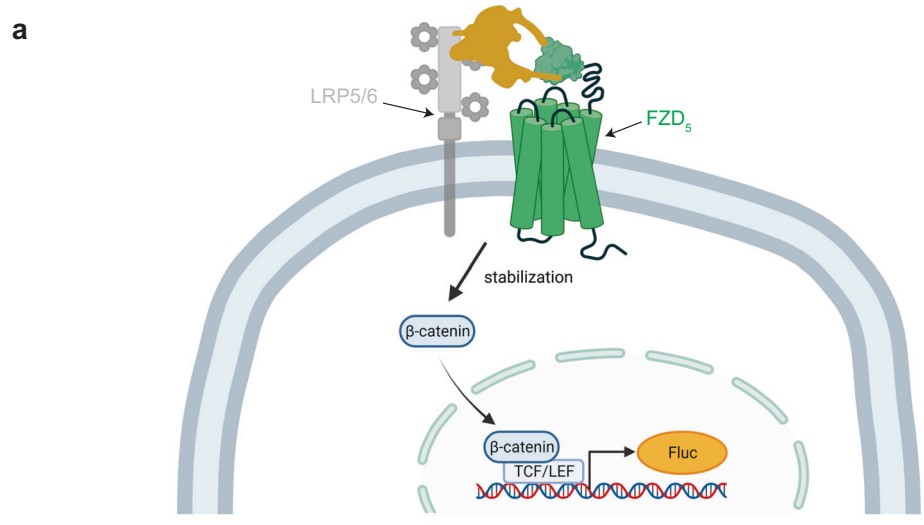

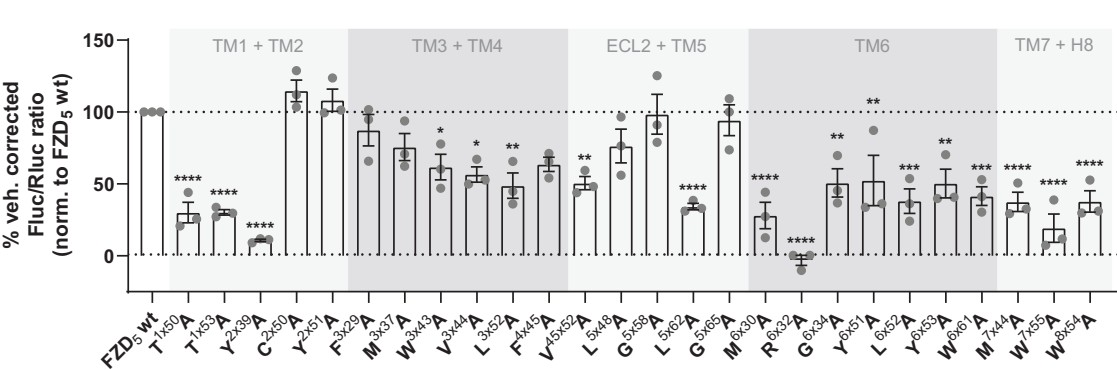

**Fig. 3 | Effects of FZD$_5$ micro-switch mutants on the WNT-3A-induced activation of β-catenin-dependent gene transcription. a** Schematic depiction of the TOP-Flash reporter gene assay; created with biorender.com. **b** TOPFlash reporter gene response in ΔFZD$_{1-10}$ HEK293T cells, transiently transfected with FZD$_5$ micro-switch mutants or wild-type FZD$_5$, upon stimulation with recombinant WNT-3A (300 ng/mL). Data show mean ± SEM of three independent experiments performed in duplicate and represent vehicle-corrected Fluc/Rluc ratios normalized to wild-type FZD$_5$ (included in every experiment). Statistical differences between the FZD$_5$ micro-switch mutants and wild-type FZD$_5$ were assessed using one-way ANOVA followed by Dunnett's post-hoc analysis. Significance levels are given as follows: *$p < 0.05$, **$p < 0.01$, ***$p < 0.001$, and ****$p < 0.0001$. TM transmembrane domain, ECL extracellular loop, H8 helix 8.

DEP and their coupling efficiency, reflected in the BRET$_{max}$ value, and were able to identify several mutants affecting one or even both parameters.

### Nucleotide-decoupled G proteins reveal stabilization of pathway selective conformations

Besides DVL-mediated pathways, FZDs are also able to initiate signaling via heterotrimeric G proteins[13,15,23–25,39,40]. Indeed, overexpression of DVL competes with FZD-G protein interaction[24,40]; furthermore, FZD coupling to heterotrimeric G proteins or DVL appears to require distinct FZD conformations, as mutation of the molecular switch residue 6×32 improves G protein coupling while it abrogates DVL binding[17,24,40]. To assess the constitutive interaction between heterotrimeric G$_q$ and FZD$_5$[13], we made use of an engineered Gα subunit, which has an elongated C terminus containing four additional alanines within the α5 helix. These Gα 4A subunits are nucleotide-decoupled and interact constitutively with cognate GPCRs[41,42]. In order to assess the ability of wild-type FZD$_5$ and the micro-switch mutants to couple constitutively to heterotrimeric G$_q$ 4A, we employed a BRET assay based on the proximity of the C-terminally Nluc-tagged receptor and Venus-tagged βγ subunits (Fig. 5a). After cotransfection of wild-type FZD$_5$-Nluc and G$_q$ 4A, we observed a robust and strong increase in the BRET response compared to conditions devoid of G$_q$ 4A, suggesting constitutive G protein coupling.

The FZD$_5$-Nluc micro-switch mutants showed a wide range of BRET responses, with some mutants behaving like wild-type FZD$_5$ and some showing reduced or even slightly increased BRET compared to wild type (Fig. 5b and Supplementary Table S4). Surprisingly, mutants, which were able to induce a strong DVL shift (L$^{5×62}$A, G$^{5×65}$A, G$^{6×34}$A, L$^{6×52}$A, M$^{7×44}$A), also displayed % BRET values in our G protein readout, which were comparable or even slightly higher (G$^{5×65}$A) than those of wild-type FZD$_5$. In contrast, mutation of the molecular switch R$^{6×32}$ or its counterpart W$^{7×55}$ to alanines resulted in maintained or even slightly reduced % BRET values compared to wild-type FZD$_5$. Several mutants however, such as C$^{2×50}$A, Y$^{2×51}$A, Y$^{6×53}$A, were completely unable to couple to G$_q$ 4A. Taken together, our data indicate that the selected micro-switch residues stabilize distinct receptor conformations, which maintain or impair efficient and constitutive G protein coupling.

### Conformational FZD$_5$ sensors reveal differences in the absence of agonist

In analogy to class A/B GPCRs, FZDs undergo conformational changes upon activation in both the extracellular regions and the transmembrane core[11,13,16,17]. In contrast to all aforementioned readouts, which reported on the activation of a distinct signaling pathway or outcome, the assessment of changes in receptor conformation represents a more global and unbiased approach to detect receptor activation. We wanted to test the impact of the mutations of state-stabilizing residues

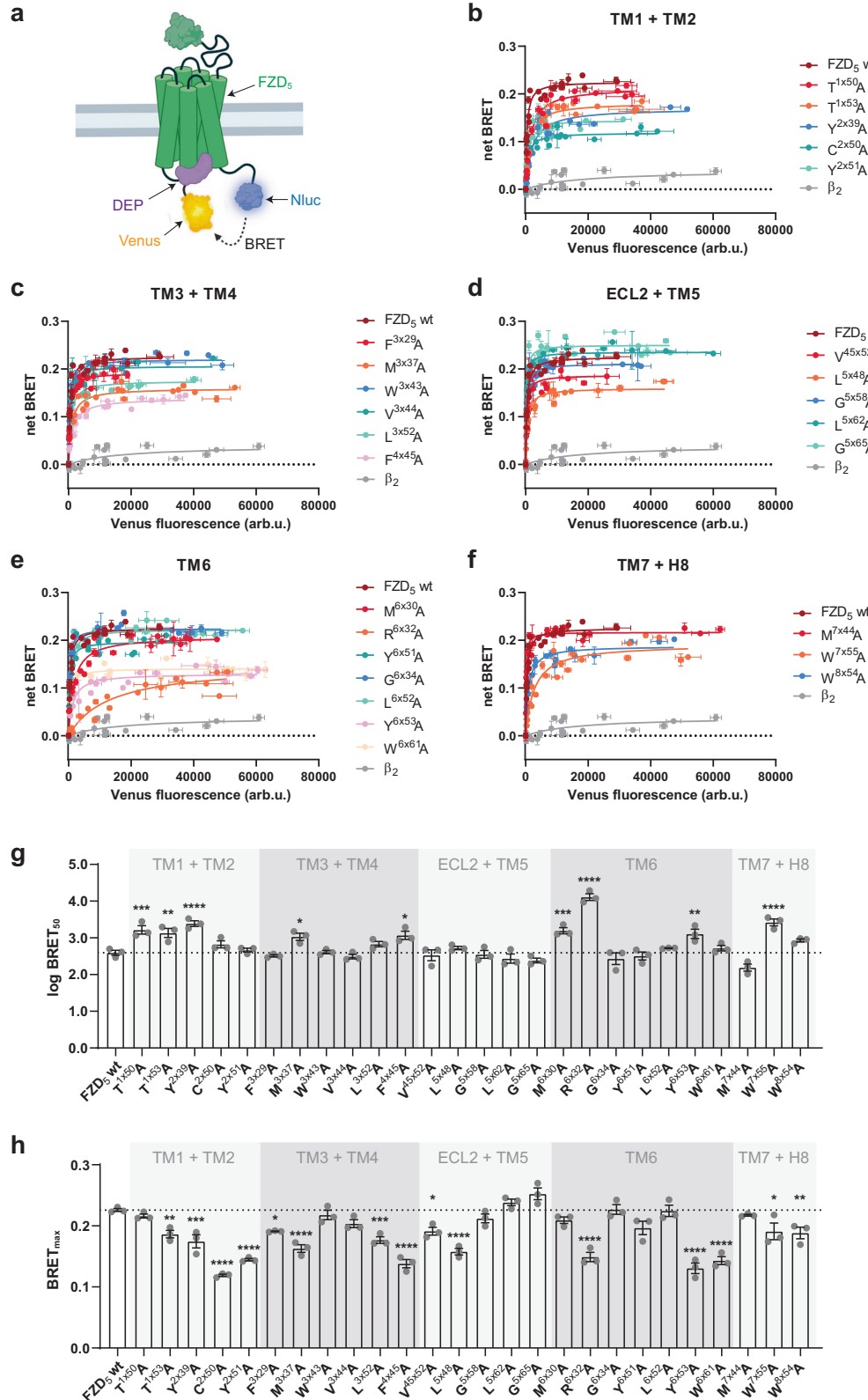

**Fig. 4 | Effects of FZD$_5$ micro-switch mutants on constitutive DEP recruitment to FZD$_5$. a** Schematic depiction of the direct BRET setup; created with biorender.com. (**b-e**) Recruitment of DEP-Venus to FZD$_5$-Nluc with mutated residues in TMs 1 and 2 (**b**), TMs 3 and 4 (**c**), ECL2 and TM5 (**d**), TM6 (**e**) or TM7 and H8 (**f**). Wild-type FZD$_5$ and the $\beta_2$-adrenergic receptor (negative control) are shown in every panel for comparison. Experiments were performed in HEK293A cells, transiently transfected with a constant amount of plasmid encoding the indicated receptor-Nluc construct and increasing amounts of plasmid encoding DEP-Venus. Data show mean ± SD of three independent experiments (data points from different experiments are superimposed) performed in duplicate. **g**, **h** log BRET$_{50}$ (**g**) and BRET$_{max}$ (**h**) values from DEP titration experiments (**b–f**) characterizing the FZD$_5$-DEP interaction. Data show mean ± SEM of three independent experiments performed in duplicate. Corresponding numerical values can be found in Supplementary Table S3. Statistical differences between the FZD$_5$-Nluc micro-switch mutants and wild-type FZD$_5$-Nluc were assessed using one-way ANOVA followed by Dunnett's post-hoc analysis. Significance levels are given as follows: *$p < 0.05$, **$p < 0.01$, ***$p < 0.001$, and ****$p < 0.0001$. TM transmembrane domain, ECL extracellular loop, H8 helix 8.

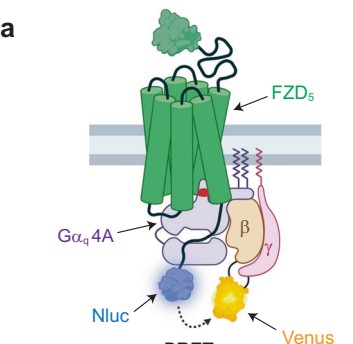

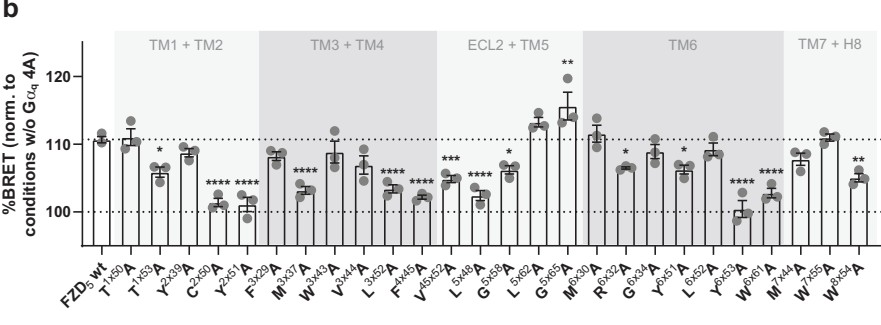

**Fig. 5 | Effects of FZD$_5$ micro-switch mutants on constitutive coupling to heterotrimeric G$_q$ 4A. a** Schematic depiction of the assay setup. The insertion of four alanines into the α5 helix of Gα$_q$ is highlighted by a red box; created with biorender.com. **b** % BRET values describing the coupling of G$_q$ 4A to FZD$_5$-Nluc. Data show mean ± SEM of three independent experiments performed in triplicate. For each experiment, values were normalized to conditions, where cells were

transfected with the same FZD$_5$-Nluc construct (wild type or mutant) but no G$_q$ 4A. Corresponding numerical values can be found in Supplementary Table S4. Statistical differences between the FZD$_5$-Nluc micro-switch mutants and wild-type FZD$_5$-Nluc were assessed using one-way ANOVA followed by Dunnett's post-hoc analysis. Significance levels are given as follows: *$p < 0.05$, **$p < 0.01$, ***$p < 0.001$, and ****$p < 0.0001$. TM transmembrane domain, ECL extracellular loop, H8 helix 8.

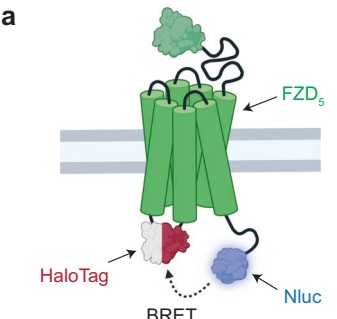

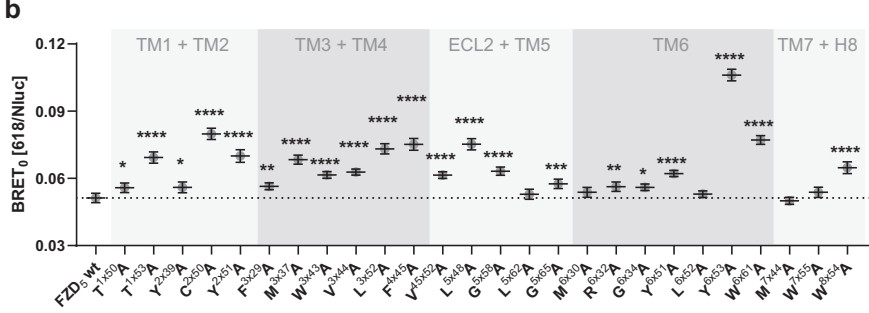

**Fig. 6 | Detection of differences in receptor conformation for the FZD$_5$ micro-switch mutants in the absence of ligands. a** Schematic depiction of the biosensor design; created with biorender.com. **b** BRET$_0$ values, obtained from experiments conducted in HEK293A cells, transiently transfected with the indicated FZD$_5$-Halo-Nluc construct (wild type or mutant). Data show best-fit values ± SD obtained from the linear fits of BRET over Nluc luminescence plots depicted in Supplementary

Fig. S6 (five independent experiments performed in sextuplicate). BRET$_0$ values can be found in Supplementary Table S5. Statistical differences between the BRET$_0$ values of the FZD$_5$-Halo-Nluc micro-switch mutants and wild-type FZD$_5$-Halo-Nluc were assessed using one-way ANOVA followed by Dunnett's post-hoc analysis. Significance levels are given as follows: *$p < 0.05$, **$p < 0.01$, ***$p < 0.001$, and ****$p < 0.0001$. TM transmembrane domain, ECL extracellular loop, H8 helix 8.

on the conformational flexibility in the transmembrane core of FZD$_5$ in the absence of ligand. Therefore, we adapted a BRET-based sensor design that has been successfully used for the generation of conformational biosensors for different class A GPCRs[11,43,44]. Changes in BRET mirror movements or conformational rearrangements within the receptor core following receptor activation. We generated FZD$_5$-Halo-Nluc by attaching the luciferase to the C terminus of the receptor (BRET donor) and inserting the self-labeling HaloTag into the third intracellular loop (ICL3) as a BRET acceptor (see Fig. 6a). The obtained conformational sensor showed basal energy transfer (see Supplementary Fig. S5a), but in contrast to a described FZD$_4$-Halo-Nluc biosensor[11], WNT-3A-stimulation of FZD$_5$-Halo-Nluc resulted in a strong and FZD-specific BRET increase (see Supplementary Fig. S5b) arguing for its functionality. Analogously to the FZD$_5$-Nluc constructs, we first mutated the state-stabilizing residues in the background of the FZD$_5$-Halo-Nluc construct to alanines and confirmed cell surface expression for all by whole-cell ELISA (see Supplementary Fig. S5c and Supplementary Table S2). Next, we recorded the basal BRET, i.e., in the absence of WNTs, for both wild-type and mutated FZD$_5$-Halo-Nluc sensors. To correct for differences in expression between the mutants, we took advantage of the previously described BRET$_0$ analysis[15,45]. Therefore, we plotted the measured BRET values as a function of the corresponding Nluc luminescence intensities and analyzed the datasets using linear regression (see Supplementary Fig. S6) yielding BRET$_0$

values as the y-intercept of the fit. A linear correlation between the datapoints was statistically confirmed for all investigated mutants (see Supplementary Table S5). Interestingly, while BRET$_0$ values for some of the FZD$_5$-Halo-Nluc micro-switch mutants were unaltered compared to wild-type FZD$_5$, most of them were significantly higher (Fig. 6b), consistent with a conformational change towards an active state. The most striking increase in BRET$_0$ was observed upon mutation of Y$^{6×53}$ to alanine, which is located in the extracellular-facing half of TM6.

### Relationship between micro-switch prediction and functional impact

We found that the mutations in expressing receptor have an effect in on average four of the seven functional endpoints evaluated in our study (Supplementary Fig. S7 and Supplementary Data 2). Strikingly, the two mutations T$^{1×53}$A and R$^{6×32}$A affected all seven studied functional endpoints and only one mutation (W$^{3×43}$A) affected a single parameter (receptor conformation). The most frequent type of functional impact was observed on the receptor conformation (21 out of 26 (81%) of surface-expressed receptors, as measured using the FZD$_5$-Halo-Nluc sensor). The lowest functional impact still affected twelve (46%) of the surface-expressed mutants and was observed in the DVL shift assay. For the DEP recruitment, the effect was more frequent on the BRET$_{max}$ rather than BRET$_{50}$ value (fifteen and nine mutants, respectively). Taken together, the GPCRdb-based design generated

mutations with frequent and multifaceted effects providing ample starting points for the functional characterization of $FZD_5$ conformation, coupling and signaling.

## Cluster analysis identifies $FZD_5$ as a DVL-prone receptor

The data originating from the characterization of numerous $FZD_5$ mutants in several independent functional assays with different parameters is complex and requires a global and unbiased way of analysis to visualize associations and correlations. For these purposes, we performed a correlation analysis of the different functional readouts (Supplementary Fig. S8) and clustered the mutations by the similarity of the experimental outcome (Fig. 7). To make the different variables comparable, we rescaled every measured variable to a range between zero and one. We found that three clusters optimally separated the observations as was determined by silhouette analysis (Fig. 7a). Wild-type $FZD_5$ clusters together with several mutants that are characterized by a broad functionality towards DVL-dependent signaling, mirrored by assays such as TOPFlash reporter gene readout, DVL electrophoretic mobility shift assay, large $BRET_{max}$ and low $BRET_{50}$ values in the DEP recruitment assay (cluster in blue). Coupling to $G_q$ 4A in this cluster appears possible albeit rather variable. The second cluster (yellow) is characterized by generally functional β-catenin-dependent signaling but a clear reduction in the assays directly focussing on DVL, such as the electrophoretic mobility shift or the weakened interaction with the DEP domain (decreased $BRET_{max}$ and increased $BRET_{50}$ values). At the same time, the $BRET_0$ analysis of receptor conformation revealed larger values indicating increased conformational flexibility. In addition, coupling to $G_q$ 4A is substantially reduced within this cluster. The third cluster (purple) comprises the previously identified molecular switch residues in TM6 ($R^{6\times32}$) and TM7 ($W^{7\times55}$) as well as other mutants that generally caused reduced (or abolished in case of $R^{6\times32}A$) TOPFlash signal and electrophoretic mobility of DVL with maintained ability to couple to $G_q$ 4A. Most importantly, this cluster contains mutants that substantially affect DEP recruitment mostly reflected by increased $BRET_{50}$ values, indicative of a reduced FZD-DEP affinity.

The outcome of the cluster analysis was basically reproduced by a principal component analysis of the dataset (Fig. 7b), where three main clusters emerged. The residues that are positioned in the corners of the analysis are $L^{5\times62}$ and $G^{5\times65}$ (blue), $R^{6\times32}$ and $W^{7\times55}$ (purple), and $C^{2\times50}$, $Y^{2\times51}$ and $Y^{6\times53}$ (yellow). To obtain an overall understanding of the spatial cluster distribution on the receptor molecule, we plotted cluster colors on the $FZD_5$ snake plot (Fig. 7c) as well as a three-dimensional $FZD_5$ model (Fig. 7d). Interestingly, the residues within each cluster are spread over the entire structure although they are functionally related. Only the purple cluster sticks out with all residues located at the intracellular side and in proximity to each other.

## MD analysis supports pathway-selective conformational dynamics of FZDs

As the principal component analysis identified state-destabilizing mutants $L^{5\times62}A$, $G^{5\times65}A$ (blue), $R^{6\times32}A$, $W^{7\times55}A$ (purple) and $Y^{2\times51}A$ and $C^{2\times50}A$ (yellow) as well as $Y^{6\times53}A$ (yellow; exceptional $BRET_0$ response) to induce the largest alterations in the response pattern compared to wild-type $FZD_5$, we set out to study the effect of these mutations on receptor dynamics. Thus, we performed MD simulations with wild-type $FZD_5$ and the alanine mutants of $L^{5\times62}$, $G^{5\times65}$, $R^{6\times32}$, $W^{7\times55}$, $Y^{2\times51}$, $C^{2\times50}$ and $Y^{6\times53}$. For each of the $FZD_5$ constructs, we ran 500 ns of an unbiased MD simulation in three independent replicas starting from a $FZD_5$ model in an inactive conformation and in the absence of an effector protein (see Supplementary Fig. S9 for backbone RMSD plots).

Overall conformational differences between wild-type $FZD_5$ and mutants as well as between mutants became evident, when aligning the representative structures of the main cluster after clustering on the backbone of the transmembrane region over all three replicas (Fig. 8a). As expected, the conformational differences were larger for the more flexible regions. However, conformational rearrangements were also observable for the transmembrane helices, e.g., at the intracellular side of TM5, TM6, and TM7 or the extracellular side of TM1 and TM6. For mutant $G^{5\times65}A$ (blue cluster), a slight outward movement of TM5 was evident (Fig. 8b, d and Supplementary Fig. S10b and d). Interestingly, the same mutation also led to a significant change of the rotamer of $W^{3\times50}$ (Fig. 8c and Supplementary Fig. S10a), which was recently proposed to be involved in an activation switch together with $G^{5\times65}$ and $M^{6\times30}$ in SMO[46]. Mutation of $W^{3\times50}$ to alanine, however, led to an almost complete loss of membrane expression when mutated in this study (Fig. 1c). A similar but less pronounced outward movement of TM5 could also be observed for $C^{2\times50}A$ (yellow cluster) while changes of the dihedral angle of $W^{3\times50}$ could also be observed for $R^{6\times32}A$, although to a smaller extent (Supplementary Fig. S10a, b and d). In the case of $L^{5\times62}A$, a movement of TM5 and TM6 towards each other could be observed (Supplementary Fig. S11). Although these changes occurred in slightly different positions on the helix and to varying extents throughout different replicas, the results clearly hint towards conformational and dynamic differences of TM5 compared to wild-type $FZD_5$ and other micro-switch mutants. While the structural rearrangements of TM5 for mutants $G^{5\times65}A$ and $L^{5\times62}A$ can be explained by a gain or loss of steric hindrance (due to the changed size of these amino acid side chains), respectively, this also indicates changed dynamics and conformational differences of the receptor, impacting the experimental outcomes, as seen above. Similarly, the rearrangement of TM7 and to a lesser extent TM6 for mutant $W^{7\times55}A$ might be explained by the loss of steric hindrance due to the drastic reduction of size of the side chain of this residue. The lower part of TM7 moved towards the core of the receptor, resulting in shorter distances to TM6 and interactions of the N-terminal part of H8 with TM2 and ICL1 (Fig. 8g, h and Supplementary Figs. S12, S15). This partial closing of the cavity at the intracellular side of the receptor could directly affect interactions with intracellular transducers.

Mutant $Y^{6\times53}A$ showed a strongly increased $BRET_0$ response in the $FZD_5$-Halo-Nluc conformational sensor compared to wild-type $FZD_5$. The MD simulation indicated that $Y^{6\times53}A$ at the extracellular side of TM6 straightens the helix in this region (Fig. 8a, e, f and Supplementary Fig. S10c), thereby reorienting ECL3 and likely also ICL3, in which the HaloTag of the conformational sensor was inserted.

For the linker domain and the loops (especially ICL3 and ECL3), different conformations could be observed for mutants and wild-type receptor (Supplementary Fig. S13). However, this was expected due to length and flexibility of these loops. In the case of the $R^{6\times32}A$ mutant, the conformation of ICL3 seemed to be rather stable, which could explain its in vitro pharmacology (Supplementary Fig. S13b). Similarly, a high degree of flexibility could be observed for H8, even resulting in insertion of the helix or parts of it into the membrane for some mutants (Supplementary Fig. S14). Although the latter behavior is likely to be an artifact, it is interesting that this could not be observed for the wild-type receptor which maintained a rather stable H8 conformation throughout all three replicas and the extent of altered flexibility differed for different mutants. Thus, our data suggest that the altered flexibility of H8 is caused by the mutations, which could be a reason for changed receptor pharmacology, as H8 is involved in effector recognition for other GPCRs[47].

In addition to the conformational changes, a clear effect of certain mutants on the frequency of observed interactions could be seen based on the interaction fingerprints of amino acid side chains (Supplementary Fig. S15 and Supplementary Data 3). These changes were also observed in regions more distant to the mutation including the

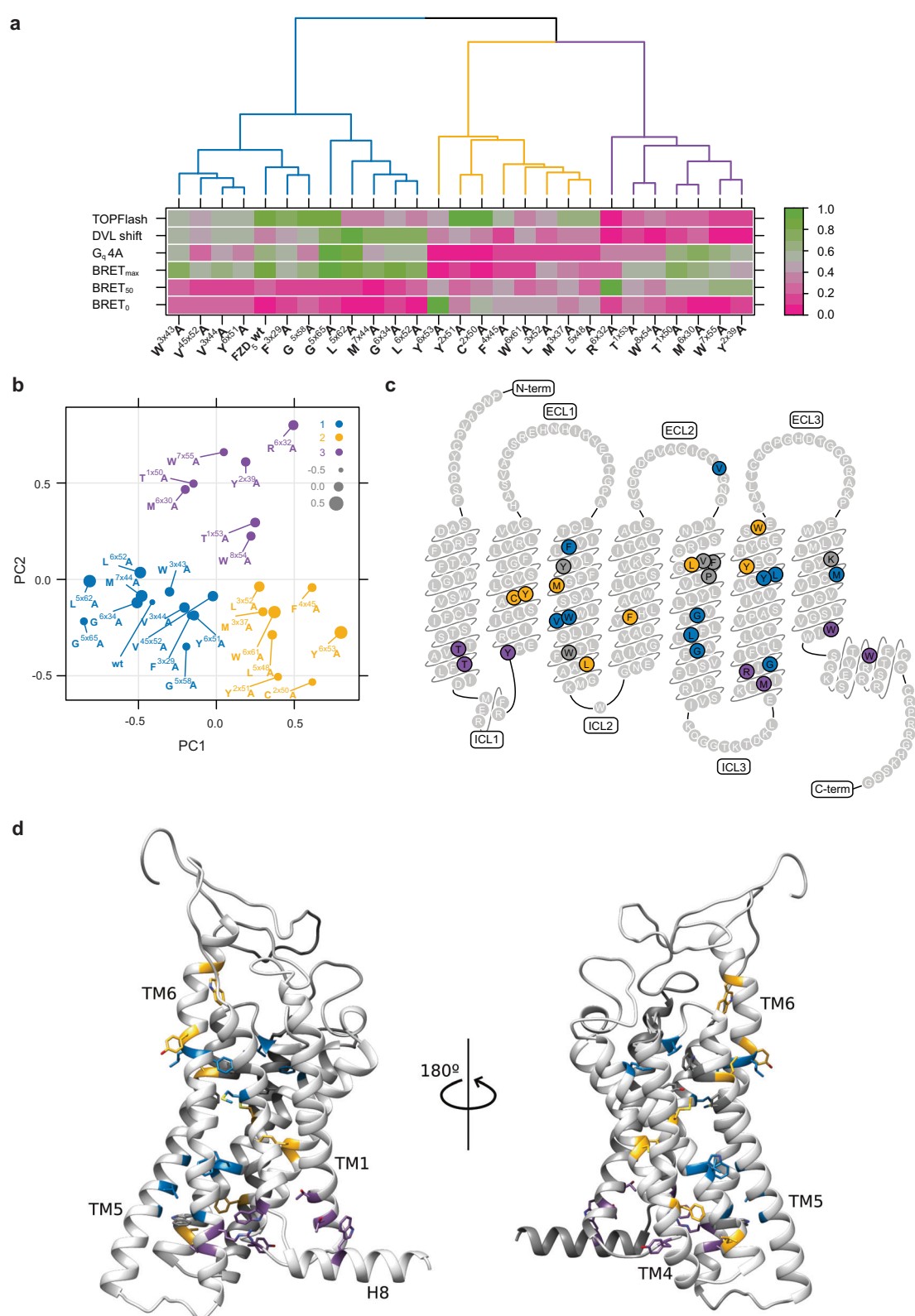

**Fig. 7 | Cluster analysis of FZD5 micro-switch mutants. a** Heat map showing combined response for the 26 FZD5 micro-switch mutants and wild-type FZD5 in five different assays (six parameters). Raw mean values for each assay were rescaled to a range between 0 and 1 for comparability. Mutants were ordered by similarity using a clustering algorithm ('ward.D2' function from R package cluster). Best clustering of mutants was obtained using three different clusters (colored in blue, yellow, and purple) according to silhouette analysis. (**b**) Principal component analysis (PCA) of mutants based on the results shown in (**a**). The color of points represents main clusters as shown in (**a**). *X*- and *Y*-axis show principal components 1 and 2, size of points encodes principal component 3. **c, d** Snake plot (**c**) and 3D model (**d**) of human FZD5 without N-and C-termini showing the 26 mutated residues colored in blue, yellow and purple according to cluster analysis in (**a**). Amino acids colored in gray represent mutants, which were not expressed. C-term C terminus, ECL extracellular loop, ICL intracellular loop, N-term N terminus, TM1-6 transmembrane domain, H8 Helix 8.

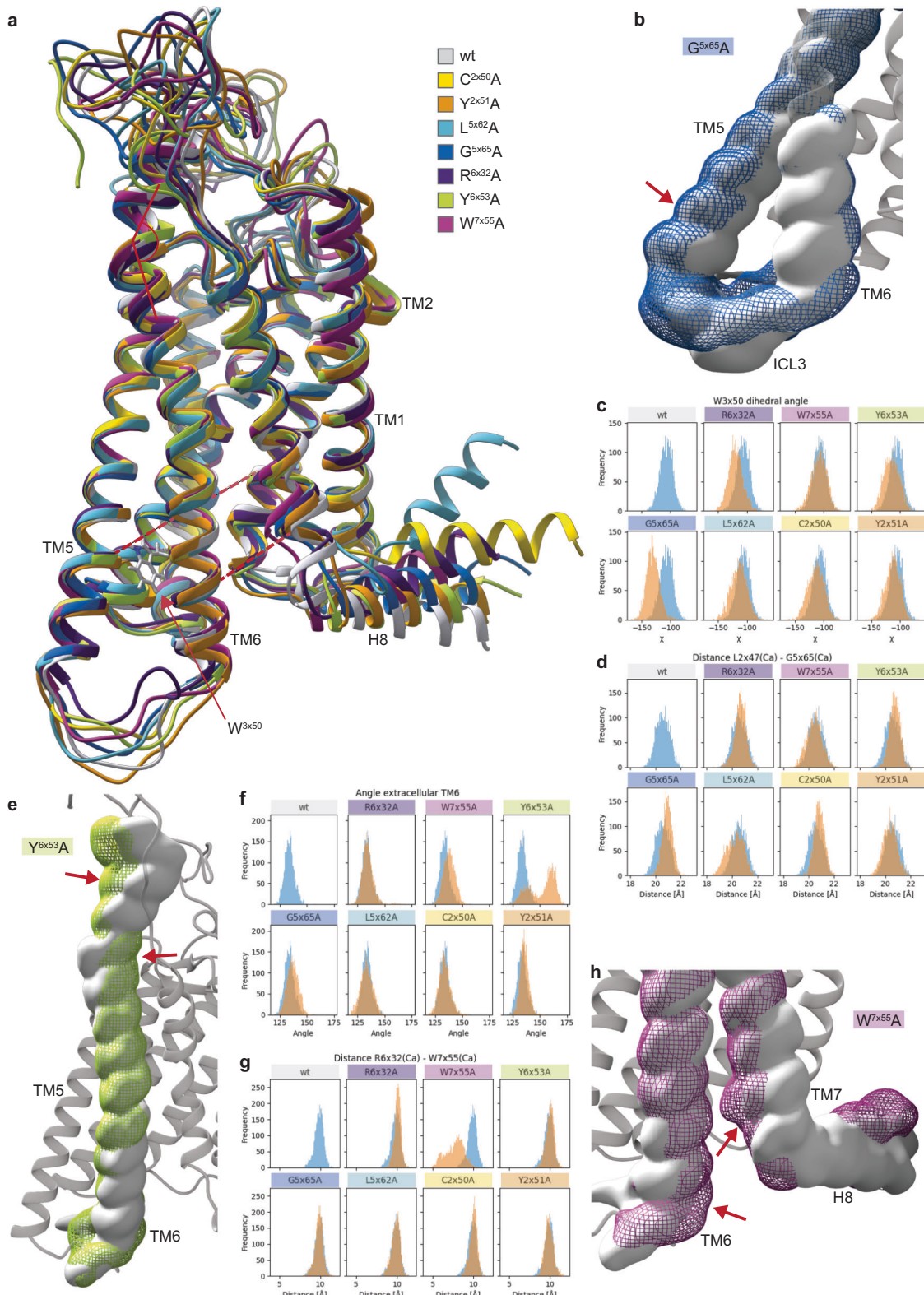

loops and the linker domain. While difficult to interpret in detail, these observations indicate that all investigated single residue mutations can induce small but global changes in interaction networks and potentially signal propagation.

In summary, the comparison of intracellular distance measurements, different measured angles, volumetric maps and cluster representatives as well as changes in interaction patterns shows that differences in specific regions are only observed for certain mutants compared to wild-type FZD$_5$ and other mutants. It should be noted that these MD simulations have their limitations since they do not sample the entire conformational landscape of the receptor and were run in the absence of intracellular transducers. However, the observations from the MD simulations strongly argue that the selected state-stabilizing residues indeed influence receptor dynamics and stabilize distinct conformational ensembles feeding into different pharmacological profiles.

**Fig. 8 | Conformational rearrangements in the mutated receptor for selected mutations as determined with MD simulations. a** Cartoon representation of the aligned representative structures of the main cluster of each system (clustered on TM region over all three replicas using each 5th frame). The arrow points to residue W$^{3×50}$ investigated in (**c**), the red dotted lines indicate the distances measured in (**d**) and (**g**) and the angle indicates the angle measured in (**f**). Cluster colors as indicated in the legend. **b**, **e**, **h** Volumetric maps of the receptor backbone atoms were calculated over the entire trajectory of all three replicas to visualize conformational differences between MD simulations of wild-type receptor and simulated mutants. **b** Volumetric map of TM5, ICL3 and lower TM6 backbone for wild-type FZD$_5$ (light gray) and G$^{5×65}$A (dark blue mesh). The slight outward shift of TM5 is highlighted (red arrow). (**c**) Histograms of occurring χ dihedral angles of residue W$^{3×50}$ calculated for each 5th frame over the concatenated trajectory of all three replicas for wild-type FZD$_5$ (blue) and indicated mutant (orange). **d**, **g** Histograms of occurring distances between Cα atoms of the indicated residues (calculated for each 5th frame of the concatenated trajectory of all three replicas) shown in blue (wild-type

FZD$_5$) and orange (indicated mutant). **d** An increase of the distance between residues 2×47 and 5×65 compared to wild type can be observed for G$^{5×65}$A and to a lesser extent for C$^{2×50}$A. **e** Volumetric map of the TM6 backbone for wild-type (light gray) and Y$^{6×53}$A (green mesh). The straightening of TM6 in the mutant compared to wild type is indicated by the red arrows. **f** Histograms of the angle of the extra-cellular portion of TM6, as indicated in (**a**), calculated for each 5th frame over the concatenated trajectory of all three replicas for wild-type (blue) and indicated mutant (orange). The generally larger observed angles for Y$^{6×53}$A confirm the straightening of the helix as described in (**e**). **g** The distance between residues 6×32 and 7×55 decreases drastically for W$^{7×55}$A compared to wild-type FZD$_5$. **h** Volumetric map of TM6, TM7, and H8 backbone for wild-type FZD$_5$ (light gray) and W$^{7×55}$A (dark magenta mesh). The movement of TM7 into the receptor core and towards TM6 and the movement of TM6 towards TM7 is indicated with red arrows. A per replica analysis of (dihedral) angle and distances for the mutants with notable deviations from wild-type FZD$_5$ in (**c**), (**d**) and (**f**) can be found in Supplementary Fig. S10 and for (**g**) in Supplementary Fig. S12.

## Impact of micro-switch mutations is transferrable to other FZD paralogs

The initial identification of potential state-stabilizing residues was based on sequence conservation in class F GPCRs. Therefore, the mutational analysis in FZD$_5$ implies that our conclusions also bear relevance for other paralogs of the FZD family. To further strengthen this assumption, we mutated the most important key residues in each cluster (note: the same residues were also investigated in the MD simulations described above) in additional FZD paralogs and tested these in some but not all of the available functional readouts. We chose FZD$_4$ and FZD$_{10}$ as two representatives of the FZD family that signal robustly in a β-catenin-dependent manner and couple to both DVL and heterotrimeric G proteins[17,24,25,36]. It became obvious that the selected residues are also stabilizing distinct receptor conformations in the case of FZD$_4$ and FZD$_{10}$, as their mutation had an impact on signaling outcomes and pathway selection. Moreover, the effect on receptor function upon mutation of these state-stabilizing residues in FZD$_4$ and FZD$_{10}$ was in agreement with the functional outcomes obtained for FZD$_5$ (see Supplementary Figs. S16–S18). For example, the parameters obtained from the DEP titration experiments (BRET$_{max}$ and BRET$_{50}$) clearly follow the same trend for the state-destabilizing mutants for each tested receptor (Supplementary Fig. S18), even though the same mutation sometimes led to significant differences in surface expression, when comparing them between the paralogs.

## Discussion

Here, we conducted a comprehensive analysis of conserved, predicted state-stabilizing residues in FZDs by performing an extensive muta-genesis and employing a complementary battery of functional read-outs for FZD-mediated signaling, transducer coupling, and conformational changes. Thereby, we were able to identify clusters of amino acids and structural networks in FZDs that impact transducer coupling differentially. This supported the concept of receptor conformation-dependent transducer selectivity of FZDs towards DVL-dependent and -independent signaling pathways[23].

Based on the comparison of two inactive and two active state receptor structures, several potentially state-stabilizing and conserved residues could be selected (see also Fig. 1a). Mutation of these residues did indeed lead to different degrees of changes in the pharmacological behavior. A larger number of distinct class F GPCRs in their inactive/ active state could have increased the share of designed mutations that ultimately influence receptor activation. However, it would not predict a larger number of state-determinant residues, as these should apply to the whole class and therefore be discerned from any pairs of inactive/active state templates.

For our experiments, we used primarily FZD$_5$ as a representative of the FZD family of GPCRs because FZD$_5$ (i) mediates WNT-induced β-catenin-dependent signaling, (ii) interacts directly with DVL via the

DEP domain, (iii) couples to heterotrimeric G$_q$ and (iv) shows con-formational dynamics. In addition, we selected FZD$_4$ and FZD$_{10}$ for a partial analysis to support our hypothesis that the impact of the state-stabilizing residues can be transferred in a class-wide fashion. By choosing these representatives of the FZD family, we were able to employ a broad range of functional readouts, which are required to obtain a more detailed understanding of structure-function relation-ships and a potential transducer selectivity of FZDs. FZD$_5$ has been extensively used to analyze various aspects of FZD-effector coupling and thus, our work merges well with previous investigations allowing direct comparisons[7,13,14,20,30,31]. Six of our initially predicted 32 micro-switch mutants were not expressed at the cell surface (Fig. 1c). While it was not sensible to further follow them up and include them in our cluster analysis, the lack of expression upon mutation itself further implies an important role for those residues in receptor integrity. Sufficient receptor surface expression is clearly decisive for receptor function and the functional analysis of the mutants created in the current study. Therefore, we chose to use low receptor expression initially as a strict exclusion criterium (Fig. 1c). In the following analysis it became obvious that (i) different functional assays show different sensitivity to receptor surface expression and that (ii) there is a yet undefined threshold for receptor surface expression—above which the impact of receptor surface expression on the functional assessment is minimal. Below this threshold level, on the other hand, the functional analysis of receptor mutants can, but does not have to be confounded necessarily (see also Supplementary Fig. S3). Nevertheless, this caveat must be considered when interpreting the data of receptor mutants with low cell surface expression compared to the wild-type receptor. In our study, this is particularly important when interpreting the mutants in the yellow cluster, as these mostly showed impaired receptor sur-face expression. For example, FZD$_5$-Nluc C$^{2×50}$A displayed a very low surface expression as well as a reduced BRET$_{max}$ value in the DEP recruitment assay, when compared to wild-type FZD$_5$-Nluc. Upon mutating the same residue in FZD$_4$-Nluc or FZD$_{10}$-Nluc, however, the receptor surface expression was equal or only slightly reduced com-pared to wild-type FZD-Nluc, whilst still displaying lower BRET$_{max}$ values than their respective wild-type receptor (Supplementary Figs. S16–S18), thereby cross-validating our findings for FZD$_5$.

The cluster analysis of our data categorized wild-type FZD$_5$ among mutants of residues that stabilize receptor conformations pre-ferentially coupling to DVL. This suggests that FZD$_5$ and FZDs in gen-eral indeed are prone to couple to DVL rather than heterotrimeric G proteins. Nevertheless, FZD coupling to heterotrimeric G proteins can be measured in many different biological systems at both endogenous and overexpressed levels of FZDs[13,15,17,23,27,48–51] as well as in purified proteins[15,52]. Thus, the important question that remains is under what circumstances the DVL-prone FZD is able to switch to coupling to heterotrimeric G proteins in a cellular context. We can only speculate

at this point whether WNTs could exert functional selectivity or whether subcellular compartmentation and local difference in access to one or the other transducer could guide coupling selection. In this context it should also be underlined that cancer mutants in the molecular switch residues $R^{6 \times 32}$ and $W^{7 \times 55}$, which for example render SMO a tumor driver, could act oncogenic in FZDs by switching FZDs from DVL to G protein coupling as previously surmised[17].

In addition to clustering the tested $FZD_5$ micro-switch mutants, we were able to correlate the experimental parameters and readouts in an unbiased manner (see Supplementary Fig. S8). For example, we only found a weak positive correlation between the DVL shift assay and the TOPFlash readout. In other words, a high ability of the receptor mutation to shift DVL does not necessarily reflect a strong activation potential in the TOPFlash readout, which is in agreement with the previously reported concept that the shift of DVL mirrors an off state of the protein[33]. These observations emphasize that both DVL shift and β-catenin signaling are complex processes, which are connected but not exclusively interdependent.

As expected, a negative correlation was obtained between the affinity of the $FZD_5$ mutants for the isolated DEP domain ($BRET_{50}$ value) and the DVL shift assay. The negativity of the correlation between $BRET_{50}$ and DVL shift is determined by the nature of the $BRET_{50}$ value describing a higher affinity with a smaller numerical value. For example, a low $BRET_{50}$ value and thus a high affinity towards DEP is reflected in a strong mobility shift of DVL, arguing that high-affinity FZD-DEP interaction is required for efficient FZD-induced DVL shift.

Based on our unbiased analysis, we were also able to identify less expected relationships between parameters. Surprisingly, the results from the $G_q$ 4A readout, mirroring constitutive G protein coupling, correlated strongly with the $BRET_{max}$ values from the DEP titration experiments. Furthermore, both $BRET_{max}$ and $G_q$ 4A values showed a strong negative correlation with the $BRET_0$ values from the $FZD_5$-Halo-Nluc conformational sensors. Almost all mutants with high $BRET_0$ values, with $Y^{6 \times 53}A$ having the largest $BRET_0$ value, were found to be located in the yellow cluster. These mutants displayed both a lack of $G_q$ 4A coupling and low $BRET_{max}$ values in DEP recruitment, while having an unaltered or only slightly decreased affinity to DEP. In contrast, receptor mutants with a lower $BRET_0$ value were distributed over the other two clusters (blue and purple) and therefore appear to be more efficient in adapting to the different transducer proteins in the absence of ligands, reflected in high $G_q$ 4A coupling and/or high $BRET_{max}$ values.

In summary, this suggests that the $FZD_5$ conformational sensors are not capable of distinguishing between coupling to DEP vs heterotrimeric G protein but are indeed reporters of the receptor's ability to efficiently accommodate either DVL or G proteins. Future work is required to resolve the structural details of distinct FZD conformations that allow specific interaction with either transducer protein. Interestingly, the concept of conformation-dependent pathway selection goes against the widely promoted signalosome concept that requires receptor oligomerization but excludes receptor conformational dynamics[7]. The systematic mutation of proposed state-stabilizing residues and the assessment with a wide array of signaling assays provided in the current work, indeed defined key residues allowing the accommodation of receptor conformations feeding into diverse signaling pathways. The MD simulations of selected $FZD_5$ micro-switch mutants additionally support the notion that alanine mutation of these key residues results in different conformational ensembles and interaction networks within the receptor. The combined findings from in vitro and in silico experiments clearly point towards conformational flexibility in FZDs as basis for signal initiation and specification. The remaining and most challenging aspect is now the integration of the two concepts to fully understand the WNT/FZD signaling system.

In summary, our study represents an important step in understanding how conformational selection defines pathway selectivity

downstream of FZDs. At the example of $FZD_5$ in an approach that allows class-wide extrapolation, we showed that FZDs prefer coupling to DVL as transducer proteins over G proteins. In line with what is known for class A and B1 GPCRs[2,29], we suggest that FZDs dynamically adopt pathway-specific receptor conformations that are stabilized by conserved micro-switches. The detailed mechanisms underlying pathway selection and the dynamics in receptor conformational ensembles remain, however, obscure. The concept presented in this work opens the door for new approaches to answer these questions, e.g., based on structural and/or computational biology. Additionally, the transducer selectivity of FZDs can further benefit future drug discovery campaigns for the development of pathway-selective FZD-targeting drugs.

## Methods

### Design of state-stabilizing mutations using GPCRdb
For each class F receptor and state (inactive/active) with a crystal or cryo-EM structure, we selected the representative template with the best resolution. Specifically, the inactive state templates included $FZD_4$ (4BD4), and SMO (4JKV), and active state structures were $FZD_7$ (7EVW), and SMO (6XBK). Inactive $FZD_5$ (6WW2) was left out due to too low resolution (3.7 Å). The two sets of inactive and active state structures were compared using two tools in GPCRdb[32,53]. Firstly, we used the Structure comparison tool [https://gpcrdb.org/structure_comparison/comparative_analysis] to identify residue-residue contacts existing in all templates for one of the states while absent in all templates for the opposite state. To ensure applicability across the whole receptor class, we required interacting residues to be conserved in at least 80% of all 11 class F GPCRs. Secondly, we used the State-affecting mutation design tool [https://gpcrdb.org/mutations/state_stabilizing] to identify individual residues for which the net sum of contact frequencies to other residues was 80% higher for one state than the other. The difference in rationale is that the latter tool assesses the net change of contacts that a residue has to other residues across the states, whereas the former tool only looks at the contacts that are 100% state-specific in the investigated templates but each of the two residues involved also might have other contacts which may counteract by stabilizing the opposite state. All selected residues were mutated to alanine to remove residue sidechains stabilizing the undesired state.

### Sequence alignment of class F receptors
The sequences of $FZD_1$ (UniProt ID: Q9UP38), $FZD_2$ (UniProt ID: Q14332), $FZD_3$ (UniProt ID: Q9NPG1), $FZD_4$ (UniProt ID: Q9ULV1), $FZD_5$ (UniProt ID: Q13467), $FZD_6$ (UniProt ID: O60353), $FZD_7$ (UniProt ID: O75084), $FZD_8$ (UniProt ID: Q9H461), $FZD_9$ (UniProt ID: O00144), $FZD_{10}$ (UniProt ID: Q9ULW2) and SMO (UniProt ID: Q99835) were aligned with Ugene and the ClustalW multiple alignment tool [https://ugene.net/]. The final class F alignment was manually edited to enable an overview of representative parts of the receptor showing the relevant micro-switch mutants.

### Cloning of FZD constructs
HA-$FZD_5$-Halo-Nluc was generated by multiple cloning steps based on HiBiT-$FZD_5$[54]. First, the HiBiT-tag was replaced with an HA-tag using the Q5 Site-Directed Mutagenesis Kit (New England Biolabs) according to the manufacturer's instruction. Nluc was amplified from Nluc-$FZD_6$[12] and attached in-frame to the C terminus of $FZD_5$ via a short, flexible linker sequence (-GSSLDGGGGS-) using XbaI and NotI restriction sites resulting in HA-$FZD_5$-Nluc (with linker). Lastly, HaloTag was amplified from $FZD_4$-Halo/Nluc[11] and inserted into the third intracellular loop after K439 (in HA-$FZD_5$-Nluc with linker) via Gibson Assembly. HA-$FZD_4$-Nluc and HA-$FZD_{10}$-Nluc were cloned by amplifying the receptor-encoding sequences from HiBiT-$FZD_4$ and HiBiT-$FZD_{10}$, respectively[54], and exchanging the $FZD_5$-encoding sequence in HA-$FZD_5$-Nluc (with linker) using BamHI and XbaI restriction sites. HA-$FZD_4$–1D4 and HA-

$FZD_{10}$-1D4 were subsequently obtained by swapping the receptor-encoding sequence in HA-$FZD_5$-1D4[16] with the respective sequences of $FZD_4$ and $FZD_{10}$ (amplified from HA-$FZD_4$-Nluc and HA-$FZD_{10}$-Nluc) via Gibson Assembly.

All mutants were created using the QuikChange Site-Directed Mutagenesis Kit (Agilent), optimized using Phusion polymerase (Thermo Fisher Scientific), except the following: For $FZD_4$ $C^{2 \times 50}A$, the receptor-encoding sequence was ordered as a gBlock (IDT) and subsequently subcloned into the HA-$FZD_4$-Nluc and HA-$FZD_4$-1D4 backbones. $FZD_{10}$ $W^{7 \times 55}A$ was generated using the Q5 Site-Directed Mutagenesis Kit according to the manufacturer's instruction. The $FZD_5$ mutants used in this study were created in the background of the synthetic HA-$FZD_5$-1D4, HA-$FZD_5$-Nluc[14] (note: no linker between $FZD_5$ and Nluc) or the HA-$FZD_5$-Halo-Nluc construct. The $FZD_4$ and $FZD_{10}$ mutants were generated in the background of the HA-$FZD_{4/10}$-1D4 or HA-$FZD_{4/10}$-Nluc constructs. All primers used in this study are listed in Supplementary Table S1 and can be found in Supplementary Data 4. All newly generated constructs were verified by sequencing (Eurofins Genomics).

### Cell culture, transfection, and treatments
Human embryonic kidney cells (HEK293A, Thermo Fisher Scientific) and $\Delta FZD_{1-10}$ HEK293T cells[35] were cultured in Dulbecco's Modified Eagle's Medium (DMEM, Hyclone) with 1% penicillin/streptomycin (Gibco, #151-40122) and 10% fetal bovine serum (FBS, Gibco) in a humidified 5% $CO_2$ incubator at 37 °C. Cell culture plastics were from Sarstedt or VWR, unless specified otherwise. All transfections were performed transiently using Lipofectamine 2000 (Invitrogen) according to the supplier's information or PEI (Alfa Aesar, linear, MW 25,000, stock solution: 1 mg/mL; PEI (μL): DNA (μg) ratio 3:1).

The absence of mycoplasma contamination was routinely confirmed by polymerase chain reaction using 5′-GGC GAA TGG GTG AGT AAC ACG-3′ (forward) and 5′-CGG ATA ACG CTT GCG ACT ATG-3′ (reverse) primers detecting 16 S ribosomal RNA of mycoplasma in the media after 2 to 3 days of cell exposure.

To assess WNT-induced effects, recombinant WNT-3A (R&D Systems, 5036-WN-010) was used. The lyophilized preparations of recombinant WNT-3A were resuspended in 0.1% bovine serum albumin (BSA, Sigma Aldrich)/Dulbecco's phosphate-buffered saline (DPBS, Hyclone) and stored at 4 to 8 °C for a maximum of 4 weeks.

### Whole-cell ELISA
For quantification of cell surface receptor expression, HEK293A cells were plated (20,000 cells/well) in transparent 96-well plates precoated with poly-D-lysine (PDL, Sigma-Aldrich). Next day, cells were transfected with 0.1 μg per well of the indicated $FZD_{4/5/10}$ constructs or pcDNA3.1 as control. After 24 h, cells were incubated with an anti-HA antibody (Abcam, ab9110, rabbit; 1:1,000) in 1% BSA/DPBS for 1 h at 4 °C. Following incubation, cells were washed five times with 0.5% BSA/DPBS and probed with a horseradish peroxidase–conjugated goat anti-rabbit antibody (Thermo Fisher Scientific, 31460, 1:2,500) in 1% BSA/DPBS for 1 h at 4 °C. The cells were washed again five times with 0.5% BSA/DPBS, and 50 μL of the peroxidase substrate 3,3′,5,5′-tetramethylbenzidine (Sigma Aldrich, T8665) were added (incubation for 30 min at room temperature). After acidification with 50 μL of 2 M HCl, the absorbance was read at 450 nm using a POLARstar Omega plate reader (BMG Labtech) or a Spark multimode plate reader (Tecan).

### Immunoblotting
The day prior transfection, HEK293A cells were seeded (100,000 cells/well) in transparent 24-well plates. The cells were transfected the next day with 0.5 μg per well of the respective $FZD_{4/5/10}$-1D4 construct (wild type or mutant) or with pcDNA3.1 as control. Cells were lysed 24 h after transfection in 2× Laemmli buffer containing 200 mM dithiothreitol (Merck). Lysates were sonicated and separated by SDS–polyacrylamide gel electrophoresis using 7.5% Mini-Protean TGX precast gels (Bio-Rad). Transfer to a polyvinylidene difluoride membrane was done with the Trans-Blot Turbo Transfer System (Bio-Rad). After transfer, membranes were incubated in 5% low-fat milk/TBS-T [25 mM Tris-HCl, 150 mM NaCl, and 0.05% Tween 20 (pH 7.6)] and subsequently in primary antibodies (diluted in the same buffer) overnight at 4 °C. The next day, the membranes were washed four times in TBS-T, incubated with goat anti-mouse or goat anti-rabbit secondary antibody conjugated to horseradish peroxidase (Thermo Fisher Scientific, anti-mouse: 31430, anti-rabbit: 31460, 1:5000, diluted in 5% low-fat milk/TBS-T), washed, and developed using Clarity Western ECL Substrate (Bio-Rad) according to the supplier's information. Primary antibodies were as follows: anti-1D4 (National Cell Culture Center, mouse; 1:1,000), anti-DVL2 (Cell Signaling Technology, 3216, rabbit; 1:1,000), and glyceraldehyde-3-phosphate dehydrogenase (GAPDH, Cell Signaling Technology, 2118, rabbit; 1:4,000).

### TOPFlash reporter gene assay
$\Delta FZD_{1-10}$ HEK293T cells (500,000 cells/mL) were transfected in suspension with 100 ng of wild-type $FZD_5$ or the respective $FZD_5$ mutant, 250 ng of the M50 Super 8x TOPFlash reporter (Addgene #12456) and 50 ng of Renilla luciferase control plasmid (pRL-TK, Promega) per mL cell suspension. Empty pcDNA3.1 was used to adjust the transfected DNA amount to 1 μg per mL cell suspension. Cells were seeded (50,000 cells/well) onto a PDL-precoated, white-wall, white-bottomed 96-well microtiter plate (Thermo Fisher Scientific). After 24 h, cells were washed once with HBSS (Hyclone) and stimulated with 300 ng/mL WNT-3A or vehicle control in serum-free DMEM containing 10 nM of the porcupine inhibitor C59 (2-[4-(2-Methylpyridin-4-yl)phenyl]-N-[4-(pyridin-3-yl)phenyl]acetamide, Abcam) to block secretion of endogenous WNTs. 24 h after stimulation, the Dual Luciferase Assay Kit (Promega, #E1910) was used for the readout. Therefore, cells were lysed with 20 μL of 1× Passive Lysis Buffer for 15 min at room temperature under shaking. 20 μL of LARII reagent were added to each well and β-catenin-dependent Fluc bioluminescence was detected using a Spark multimode microplate reader (Tecan, 550-620 nm, integration time: 2 s). Next, 20 μL of 1× Stop-and-Glo reagent were added per well and Rluc bioluminescence was recorded (445-530 nm, integration time: 2 s) to account for differences in transfection efficiency. Experiments were performed using a Spark microplate reader (Tecan).

### BRET-based DEP recruitment assay
HEK293A cells (300,000 cells/mL) were transfected in suspension with 20 ng (per mL cell suspension) of wild-type $FZD_{4/5/10}$-Nluc, the respective $FZD_{4/5/10}$-Nluc mutant or $\beta_2$-Nluc and varying amounts of DEP-Venus[14]. Empty pcDNA3.1 was used to adjust the transfected DNA amount to 1 μg per mL cell suspension. Cells were seeded (30,000 cells/well) onto PDL-precoated, black-walled, black-bottomed 96-well microtiter plates (Greiner BioOne). Two days after transfection, cells were washed once with HBSS and kept in 90 μL of HBSS. Venus fluorescence was read three times using a TECAN Spark microplate reader (excitation: 485/20 nm, emission: 535/25 nm). Next, 10 μL of coelenterazine h (Biosynth; final concentration: 5 μM) were added per well. After 6 min of incubation at 37 °C in the dark, the BRET ratio was recorded three times. All experiments were conducted at 37 °C using a Spark multimode microplate reader (Tecan). Bioluminescence intensity originating from Nluc was recorded between 445 and 485 nm, whereas acceptor emission (Venus) was detected between 520 and 560 nm. Both light emissions were recorded with the same integration time of 200 ms (for $FZD_4$ and $FZD_{10}$) or 300 ms (for $FZD_5$).

### BRET-based G 4A coupling assay
HEK293A cells (300,000 cells/mL) were transfected in suspension with FZD-Nluc (wild-type or mutant, 10 ng for $FZD_4$-Nluc and $FZD_{10}$-Nluc, 20 ng for $FZD_5$-Nluc), 250 ng of Venus(1-155)-$\beta_1$, 250 ng of Venus(156-239)-$\gamma_2$ and 400 ng of $G_{q/12/13}$ 4A[55] or pcDNA3.1 per mL cell suspension.

The transfected DNA amount was adjusted to 1 µg per mL cell suspension with empty pcDNA3.1. Transfected cells were seeded (30,000 cells/well) onto PDL-precoated, white-wall, white-bottomed 96-well microtiter plates (Thermo Fisher Scientific). Two days after transfection, cells were washed once with HBSS and kept in 90 µL of HBSS. Next, 10 µL of coelenterazine h (final concentration: 5 µM) were added to each well. The plate was incubated at 37 °C for 6 min in the dark and the BRET ratios were recorded three times. All experiments were performed at 37 °C using a Spark multimode microplate reader (Tecan) with the following settings: Donor emission (Nluc) was detected between 445 and 485 nm and acceptor emission (Venus) between 520 and 560 nm. Integration times were set to 100 ms for recording of both emissions.

## FZD₅-Halo-Nluc BRET measurements

For BRET measurements with the FZD₅-Halo conformational sensors, HEK293A cells (300,000 cells/mL) were transiently transfected with 20 ng of wild-type FZD₅-Halo-Nluc or the respective FZD₅-Halo-Nluc mutant per mL of cell suspension. Empty pcDNA3.1 was used to adjust the DNA amount to 1 µg per mL cell suspension. Cells were seeded (30,000 cells/well) onto PDL-precoated white-wall, white-bottomed 96-well microtiter plates (Thermo Fisher Scientific). After 24 h, 10 nM C59 and 50 nM HaloTag NanoBRET 618 ligand (#N1661, Promega) were added to the cells. Next day, cells were washed once with HBSS and incubated with 100 µL of a 1/1,000 dilution of furimazine stock solution (#N1661, Promega) in 0.1% BSA/HBSS. After 5 min of incubation at 37 °C, the basal BRET ratio was recorded for about 60 min.

For WNT-3A-induced kinetic measurements, cells were washed once with HBSS and incubated with 90 µL of a 1/1,000 dilution of furimazine stock solution in 0.1% BSA/HBSS. After 5 min of incubation at 37 °C, the basal BRET ratio was measured in three consecutive reads, after which 10 µL of a WNT-3A solution (in 0.1% BSA/HBSS, final concentration: 1 µg/mL) or vehicle control were applied per well. Solutions were prepared in Sigmacote (Sigma Aldrich, SL2)−precoated transparent 96-well plates to avoid adsorption of recombinant proteins to plastic surfaces. Subsequently, the BRET ratio was recorded for an additional 45 to 60 min.

All experiments were conducted at 37 °C using a Spark multimode plate reader (Tecan). The emission intensity of the donor (Nluc) was detected between 445 and 485 nm, acceptor emission intensity (HaloTag NanoBRET 618 ligand) between 595 and 650 nm. An integration time of 50 ms was applied for the recording of both emissions.

## Correlation analysis and data availability

Analysis for clustering and plotting of data from different functional assays was performed using the R programming language (version 4.2.0). Raw mean values from all assays were rescaled to a range of 0 to 1 for comparability and then clustered by similarity using 'hclust' function from package 'cluster' with method 'ward.D2'. The optimal cluster number (best separation) was obtained by silhouette analysis from package 'cluster'. Principal component analysis was performed using function 'prcomp' from package 'stats'. Snakeplot templates in SVG format were obtained from https://gpcrdb.org/, and modified in R using the custom 'fluctuator' package [https://github.com/m-jahn/fluctuator].

## Receptor modeling and molecular dynamics simulations

To model the FZD₅ structure, the sequence between residues G180-C539 of human FZD₅ was used, thereby removing the CRD as well as the C-tail of the receptor while maintaining important residues of the linker domain and the entire helix 8. The model was then built with MODELLER 10.1[56] using the templates listed in Supplementary Table S6. Importantly, all used templates were in an inactive conformation resulting in a FZD₅ model in an inactive conformation. During modeling, five cysteine disulfide bonds at the extracellular side

(linker domain, ECL1 and ECL3) were pre-defined based on structural knowledge and cysteine locations in the sequence (C192-C218, C222-C298, C315-C390, C182-C486, C190-C484). Of the five generated models the two with the best DOPE scores were inspected closer since their scores were very similar. Finally, the model with the second-best DOPE score was selected for further experiments since the molecular switch (R⁶ˣ³² and W⁷ˣ⁵⁵) was closed (i.e., residues were stacked) in this model[17]. The model was inspected visually to rule out artificially deformed backbone stretches and subsequently used further without any additional minimization.

The simulation system was set up and input files were generated using the CHARMM-GUI bilayer builder[57,58]. The FZD₅ model was oriented by aligning to the entry for PDB ID 6WW2[7] in the Orientations of Proteins in Membranes database using UCSF Chimera (v.1.13)[59,60]. Histidine protonation states were assigned manually based on the local environment. All pre-defined disulfide bonds from the modeling in the extracellular parts of the receptor were assigned accordingly. The termini were patched by acetylation and methylamidation. The receptor was placed in a palmitoyl-oleoyl-phosphatidylcholine (POPC) bilayer using a hexagonal box. The system was then solvated using TIP3P water and Na⁺ and Cl⁻ ions were added up to a concentration of 0.15 M to neutralize the system. For the mutated receptor systems (L⁵ˣ⁶²A, G⁵ˣ⁶⁵A, R⁶ˣ³²A, W⁷ˣ⁵⁵A, Y²ˣ⁵¹A, C²ˣ⁵⁰A and Y⁶ˣ⁵³A) the Ala point mutation was introduced to the FZD₅ model using UCSF Chimera (v.1.13) and the Dunbrack-Rotamer library[59,61]. The simulation systems for the mutated receptors were then set up as described above using the CHARMM-GUI bilayer builder.

Each system was equilibrated, and production runs were performed in three independent replicas using the CHARMM36 and CHARMM36m force field in GROMACS 2021.3[62–64]. After a short minimization over up to 3500 steps using steepest decent, velocities were assigned randomly based on the Maxwell distribution at 310 K at the beginning of the equilibration. During the equilibration, positional restraints on the atoms were removed stepwise over a total of six steps. After an initial equilibration in an NVT ensemble over a total of 250 ps, the systems were further equilibrated in an NPT ensemble for a total of 11.75 ns. During the NPT equilibration steps, the temperature of 310 K was maintained using a Berendsen thermostat and the pressure of 1 bar was maintained using a Berendsen barostat[65].

Production runs were performed without restraints for 500 ns per replica using time steps of 2 fs. The temperature of 310 K and the pressure of 1 bar were contained using the Nosé-Hoover thermostat[66] and the Parrinello-Rahman barostat[67,68], respectively. Bonds to hydrogen atoms were constrained using the LINCS algorithm[69]. Long-range electrostatic interactions were calculated using the Particle-Mesh Ewald (PME) algorithm[70] with a grid-spacing of 1 Å while long-range non-bonded interactions were cut off using a smooth force-switch between 10 and 12 Å.

After finishing the production runs, the trajectories were centered using GROMACS 2021.3[64] and then post-processed using AmberTools 18 CPPTRAJ[71]. The trajectories of all replicas of the same system were concatenated for all analyses, unless it is specifically stated that the analysis was conducted per replica. All measurements such as RMSD, dihedral angles, distances and helix angles as well as volumetric maps of the backbone atoms were determined using AmberTools 18 CPPTRAJ[71]. Interactions were determined, frequencies calculated, and interaction fingerprint maps plotted using getcontacts [https://getcontacts.github.io/]. Trajectories were visualized in VMD[72], while volume maps were visualized and images created using UCSF Chimera (v.1.13)[59,73]. All trajectories are deposited on GPCRmd[74].

## Data and statistical analysis

For analyzing the surface expression of the micro-switch mutants and pcDNA3.1-transfected cells, the mean absorbance values were normalized to the respective wild-type FZD construct, which was set to 100%.

The shown immunoblot data for FZD$_5$ are representative of six independent experiments. The FZD$_5$-induced electrophoretic mobility shift of DVL2 was quantified by densitometry of the two predominant DVL2 bands (lower band as basal DVL2, and every higher than lower band as shifted DVL2) using ImageLab (Bio-Rad). After background subtraction, the ratios of the values shifted (PS-) DVL2/basal DVL2 were calculated. Those ratios were normalized to wild-type FZD$_5$, which was set to 100% and included on each single blot.

TOPFlash ratios were calculated by dividing the Fluc emission (β-catenin-dependent transcriptional activity) by the Rluc emission (indicator for transfection efficiency). For each mutant, the TOPFlash ratios in the presence of WNT-3A were divided by the respective TOPFlash ratios obtained after the addition of vehicle control to determine the increase over baseline. The obtained values were then min-max normalized to wild-type FZD$_{4/5/10}$ (included in every experiment) and 1 (i.e., no increase over baseline) to correct for experimental variability.

BRET ratios were defined as the acceptor emission (Venus or HaloTag NanoBRET 618 ligand) over the donor emission originating from Nluc. For DEP titrations, both the fluorescence (external excitation before substrate addition) and BRET ratios (measured after substrate addition) were recorded at least three times and averaged. For every titration, a control transfection containing the same amount of plasmid encoding FZD$_{4/5/10}$-Nluc (wild-type or mutant) but no DEP-Venus plasmid was included to correct for basal fluorescence and BRET, resulting in net BRET values for the latter. For the G$_{q/12/13}$ 4A experiments, BRET ratios were recorded three times and averaged. For each experiment and for every FZD$_{4/5/10}$-Nluc variant (wild-type or mutant), the BRET ratios measured in the presence of G$_{q/12/13}$ 4A were normalized to the BRET ratios determined for the same receptor-Nluc construct in the absence of G$_{q/12/13}$ 4A.

For the FZD$_5$-Halo-Nluc conformational sensors, BRET over Nluc luminescence plots were generated by using data from three different timepoints of each of the five independent experiments.

Raw data from the plate reader were obtained as Microsoft Excel spreadsheet (.xlsx format) and analyzed using GraphPad Prism 9.0 (GraphPad, San Diego, CA, USA). Net BRET datasets from DEP titrations were fitted using a one site-specific binding equation yielding BRET$_{50}$ and BRET$_{max}$ values. In order to be able to indicate symmetrical errors (normal distribution), BRET$_{50}$ values were subsequently log normalized. Datasets from BRET over luminescence plots (FZD$_5$-Halo-Nluc sensors) were analyzed by linear regression. A potential deviation from linearity was tested with a runs test ($p < 0.05$ was considered significant). BRET$_0$ values represent the best-fit value for the $y$-intercept of the linear fit and are given with the standard error of the fit. Wherever applicable, data were tested for statistically significant differences (when $p < 0.05$) using one-way analysis of variance (ANOVA) followed by Tukey's or Dunnett's post-hoc analysis or Fisher's least significance difference (LSD) test as indicated in the figure legends.

## Data availability

Data supporting the findings of this manuscript are available from the corresponding author upon request. A reporting summary for this article is available. MD simulations are deposited in the GPCRmd database and are accessible via the Dynamics IDs 1260, 1262, 1263, 1264, 1265, 1266, 1267, and 1268. Expression vectors used and created for this work can be obtained from corresponding author. Source data for all experiments are attached as a Source Data File and are also available on Github [https://github.com/m-jahn/R-notebook-microswitches] and Zenodo [https://doi.org/10.5281/zenodo.8088056]. Source data are provided with this paper.

## Code availability

The raw data from functional assays and the associated R code is publicly available on Github [https://github.com/m-jahn/R-notebook-

microswitches] and Zenodo [https://doi.org/10.5281/zenodo.8088056]. All steps of the analysis are documented in an R notebook available in the same repository.

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

## Acknowledgements

The authors thank Benoit Vanhollebeke for the ΔFZD$_{1-10}$ HEK293 cells and Thomas Sakmar for the synthetic FZD$_5$. Thanks to Hannes Schihada, who was essential in the initial phase of the project conceptualization, and Ainoleena Turku for comments on the manuscript. The work was supported by grants from Karolinska Institutet, the Swedish Research Council (G.S.: 2019-01190), the Swedish Cancer Society (G.S.: 20 1102 PjF; P.K.: 20 0264P), the Novo Nordisk Foundation (G.S.: NNF20OC0063168, NNF21OC0070008, NFF22OC0078104), The Wenner-Gren Foundations (J.B.: UDP2021-0029), The German Research Foundation (L.G.: 504098926; M.K.J.: KO 5463/1-1, M.M.S.: 470002134). This project has received funding from the Innovative Medicines Initiative 2 Joint Undertaking (JU) under grant agreement No 875510. The JU receives support from the European Union's Horizon 2020 research and innovation programme and EFPIA and Ontario Institute for Cancer Research, Royal Institution for the Advancement of Learning McGill University, Kungliga Tekniska Högskolan, Diamond Light Source Limited. The computations were enabled by resources provided by the National Academic Infrastructure for Supercomputing in Sweden (NAISS) and the Swedish National Infrastructure for Computing (SNIC) at National Supercomputer Centre (NSC) in Linköping and KTH Royal Institute of Technology (PDC) in Stockholm (G.S.: SNIC 2021/5-490, M.M.S.: SNIC 2022/22-558) partially funded by the Swedish Research Council through grant agreements no. 2022-06725 and no. 2018-05973. D.E.G. acknowledges funding from the Novo Nordisk Foundation (NNF18OC0031226) and Lundbeck Foundation (R313-2019-526). D.E.G., G.S., and M.M.S. are members of the COST Action ERNEST (CA18133), supported by COST (European Cooperation in Science and Technology, www.cost.eu).

## Author contributions

L.G., M.K.J., and G.S. conceived and designed the study. D.E.G. designed the predicted state-stabilizing mutations. L.G., M.K.J., P.K., and J.B. performed the wet lab experiments. M.J. performed correlation analysis. M.M.S. performed and analyzed molecular modeling and MD simulations. N.A.L. provided valuable tools. L.G., M.K.J., M.M.S., and G.S. designed and prepared the figures. L.G., M.K.J., M.M.S., and G.S. wrote the manuscript. D.E.G., N.A.L. commented and contributed to the manuscript writing. G.S. supervised and coordinated the project.

## Funding

## Competing interests

The authors declare no competing interests.
