## [Peer Review File · Nature Communications]

REVIEWER COMMENTS

Reviewer #1 (Remarks to the Author):

This is a well written and interesting manuscript that attempts to identify key microswitches in the structure of Frizzled 5 (FZD5) that are conserved state-stabilizing residues that might be responsible for determining pathway selection between coupling to Disheveled (DVL) and heterotrimeric G proteins (in this case Gq). The work builds on recent work from the author's lab that shows that disease-linked cancer mutations in key molecular switch residues in SMO (R6x32 and W7x55) are drivers for cancer and can switch coupling to G-proteins. The authors have identified key state-affecting mutations in FZD5 by comparing active and inactive structures of SMO, FZD4 and FZD7. They have then gone on to assess the impact of alanine mutations in these residues on a panel of sophisticated readouts that include DVL mobility shifts, reporter gene measurements of WNT-3A stimulated beta-catenin signalling as well as BRET assays to monitor DEP and Gq-4A recruitment and BRET-based intramolecular conformational changes. Some nice cluster analysis and molecular dynamics have then been performed to group the targeted residues into three clear groups. The main aim of the study was to understand how conformational selection can determine pathway selectivity downstream of FZDs. The main conclusion is that FZDs prefer coupling to DVL as a transducer.

The one concern I have is that the authors have predominantly applied this analysis to constitutive receptor coupling/signalling in the absence of agonist. The only situation in which agonist-stimulation was applied was in the case of the reporter gene response that measured beta-catenin-dependent gene transcription. As a consequence, one is left with the question of whether WNT-3A activation changes the coupling to DEP and Gq-alpha and alters the conformational changes observed with the intramolecular BRET sensor? Surely this is worth adding into this manuscript to get a clear view of the role of the identified microswitches in mediating agonist-induced pathway selectivity? It would also be worth knowing the actual level of expression of FZD (in the different assays; compared to endogenous expression levels) since this can obviously induce constitutive coupling at high receptor expression levels. I accept that the authors have done some good control experiments to ensure that relative cell surface expression has been monitored for each alanine mutant in each experimental set-up.

Reviewer #2 (Remarks to the Author):

In the manuscript "Pathway selectivity in Frizzleds is achieved by conserved micro-switches defining pathway-determining, active conformations" the authors present a broad and thorough study of potential key microswitch residues identified in FZD5 via comparative analysis with known structures of GPCRs. In the study, the authors perform an alanine scan targeting functionality of these residues and assess outcomes in an extensive panel of assays reflecting various aspects of receptor behavior in Dvl-, beta-catenin- and G protein-related readouts. The accumulated data included DVL shift and interaction with DEP domain, TopFlash readout, Gq 4A-based G protein coupling readout and intrinsic conformational sensor based on HaloTag insertion in the ICL3. These results were summarized and clustered yielding individual signaling groups which were proposed to delineate structural ensembles responsible for receptor coupling to distinct pathways. Molecular dynamics simulation of receptor model was chosen as a method to investigate in more details which exact structural rearrangements might guide these preferences. Overall, the work is broad, technically sound and beyond doubt brings significantly further our understanding of the structural determinants of the functioning of the Frizzleds. However, several major and minor concerns must be addressed by the authors:

Major concerns:

- The authors begin with removal from analysis the mutants which do not show statistically significant presence at the cell surface. However, the method they have chosen (ELISA) has a clear sensitivity limit evident even on the Fig.1c and can be estimated to reliably reports surface levels in the range of >10% of that of WT protein. However, from their further data it is clear that there is little, if any, correlation between surface expression and activity in any of the assays. Therefore, even 1% of the surface expression might be sufficient to obtain measurable readouts, and thus claim that these mutants are not expressed at the cell surface might be at least misleading. Possibly, exclusion of

these mutants from analysis is also not justified, though, admittedly, their aberrantly low expression levels are of concern and most likely they shouldn't be included in the analysis. Nevertheless, I suggest that the authors should perform at least one, better two functional assays of different nature, e.g. TopFlash and Gq 4A, to re-evaluate signaling potential of these mutants and reconsider their conclusions on them.

- It is unclear how the TopFlash data, which is obtained upon stimulation by Wnt3a is combinable with the rest of the approaches which rely on ligand-free activity of the receptors. It would be logical to make a separate data set with tests of Wnt3a activity on the responsive Halo-tagged construct and of Wnt5a on Gq 4A which was shown to function in the authors' previous works. Such dataset, together with TopFlash data can be treated separately to analyze distinct roles of these microswitches in mutant-intrinsic and Wnt-responsive activities

- The authors' claim that they uncovered an array of microswitches universally applicable across the entire FZD family is based solely on 80% conservation cutoff. To maintain such claim they should introduce some representative mutations, e.g. the ones used in MD simulations, in at least one or better several other FZDs and demonstrate similar behavior. Admittedly, to a degree, this has been already done for residues R/K6.32 and W7.55 on FZD6 in the other work by the authors, but it clearly falls short of universality in context of new findings described in this work. Otherwise the "Frizzleds" in the title must be replaced with "Frizzled-5" and corresponding statements across the manuscript should be softened.

- The part describing molecular dynamics analysis lacks coherence. It is stated that 2 representatives were taken from each cluster, but then all the details of resulting changes are discussed either individually or independently of clustering and no global conclusion is drawn. The data on the Supplementary Fig. 7 is underdescribed as there are clearly several interactions, which appear to be cluster-specific.

- It is stated that for MD analysis 3 independent replicas were made for each mutant, but out of all the figures presented by authors this is only shown in Supplementary Fig.S6. The RMSD is characterized by a strong "saturating" effect because it measures differences from the starting structure using quadratic distances. This means that it rapidly becomes high and almost flat, not because the protein no longer changes, but because the protein continues to change populating conformations that are different from each other but are equally "distant" from the initial structure. Therefore, we cannot say that all 3 repeats are very similar using RMSD plots, but authors do not provide any other proper assessment of how reproducible were the results from independent replicas. The data shown on the Fig. 8 and Supplementary Figure S7 should be reassessed in this respect: instead of showing the averaged or pooled data, the data from independent replicas should be treated essentially as independent repeats and the difference assessed statistically. A relevant minor issue here - MD simulation results should be clearly annotated with how the replicate data was taken into account, at least in the figure legends and materials and methods.

- The authors provide a detailed discussion of the structural effects of the mutations, with major rearrangements occurring in TM5 and TM6, which appears similar to class A and B GPCR activation and activated state of Smo; however, there seem to be no specific effects of TM3. It would be useful if authors add some discussion of similarities and differences between mechanisms of FZD5 and GPCRs of other classes.

Minor concerns:

- numeration in the text is not uniform. The authors sometime use "." and sometimes "x" as a separator for their system of residue numeration, and sometimes use simply their number in the sequence. The supplementary Figure S7 has conversion between two systems in the legend which is inconvenient, and only for the mutants, not for interaction pairs

- colors on the figure 8 are given as text, which makes the figure difficult to read. The authors should put a small legend with the color coding of volumetric representations. Color coding of the clusters should be shown for residues on Fig. 8b, d, g.

- the authors vaguely state that all selected residues are at least 80% conserved across the FZD family. It would be useful to make a supplementary figure with alignment of FZD family highlighting mutated positions

Reviewer #3 (Remarks to the Author):

Grätz and Kowalski-Jahn et al. investigate conserved state-stabilizing residues in the FZD family of receptors and report their role in pathway selectivity. The FZD5 has served as a representative receptor to experimentally assess a panel of functional signaling readouts including FZD-mediated WNT/b-catenin signaling, FZD-DVL interaction, FZD-G protein coupling, and conformational dynamics. The study is complemented with molecular dynamics simulations to study the molecular impact of validated mutations on receptor structure/dynamics at atomistic resolution. Based on this data, the authors test also the hypothesis of whether distinct receptor conformations determine downstream signaling for FZD5 as a representative of the FZD family. Ultimately, a clustering approach reveals unexpected correlations between different functional outcomes.

Major comments:

1. In a first step, authors predict state-stabilizing residues for mutagenesis of FZD5. Can the authors discuss and comment on the quality of this prediction related to the hit rate of identified residues with a functional impact? Moreover, does the experimentally observed functional impact correlate to the number of altered contacts when comparing different receptor states? Of specific interest are also residues that have been predicted to be state stabilizing but did not show any impact upon mutation. What is the percentage of those residues? All in all, I would encourage the authors to dedicate a whole section to this relevant topic.

2. Section "Conformational FZD5 sensors reveal differences in absence of agonist". I acknowledge the effort of the authors to investigate also conformational changes. However, at the current state, this section is not very conclusive. Unfortunately, the experimental setup allows only detecting if a conformational change occurs but without further specifications. Considering the differential impact on the functional outcome upon mutation, it would have been interesting to learn what type of helix movements correspond to a distinct functional outcome.

For instance, the authors write "Interestingly, while BRET0 values, most of them were significantly higher (Fig. 6b), consistent with a conformational change towards an active state." How does this go along with the observation that numerous mutants with "conformational changes towards an active state" show reduced Gq 4A coupling in Figure 5a.

3. Section "MD analysis supports pathway-selective conformational dynamics of FZDs". A better representation of the data would improve the understanding of the structural data. For instance, for the G5x65A mutant, authors show a TM5 movement in Figure 8a which is quantified as distance I2x44-G5x65 in Figure 8d. This distance distribution is informative, similar to the other distribution plots in Figure 8. However, relevant information is lost by showing only the occupancy maps of TM5 and TM6. I propose depicting instead a representative structure for the average TM5-TM6 distance in cartoon superimposing the WT to the mutant. Such a depiction also allows showing (i) the side chains of the mutant and WT, (ii) the monitored TM5-TM6 distance as well as (iii) other correlated residues such as W3x50 in the same image. Such representation would also improve the understanding of the TM5-TM6 distance in Figure 8f and the related re-orientation of ICL3 as well as the TM6-TM7 positioning in Figure 8h.

Minor comments:

4. In the introduction a different numbering scheme (e.g., W7.55) is used compared to the results section (e.g., W7x55)

5. Please add in Figure S7 the residue numbering scheme to facilitate the reading.

6. In the structure comparison tool of GPCRdb, when comparing "Two sets of structures", the

structure selection remains "None selected" despite having selected structures. Please correct this error.

Rebuttal letter

In the text below we address the reviewer's comments, which were very helpful to guide the revision efforts. Our additions are in **green**. In case new data are presented in the manuscript or the supplementary information, this is emphasized in **bold letters**.

REVIEWER COMMENTS

Reviewer #1 (Remarks to the Author):

This is a well written and interesting manuscript that attempts to identify key microswitches in the structure of Frizzled 5 (FZD5) that are conserved state-stabilizing residues that might be responsible for determining pathway selection between coupling to Disheveled (DVL) and heterotrimeric G proteins (in this case Gq). The work builds on recent work from the author's lab that shows that disease-linked cancer mutations in key molecular switch residues in SMO (R6x32 and W7x55) are drivers for cancer and can switch coupling to G-proteins. The authors have identified key state-affecting mutations in FZD5 by comparing active and inactive structures of SMO, FZD4 and FZD7. They have then gone on to assess the impact of alanine mutations in these residues on a panel of sophisticated readouts that include DVL mobility shifts, reporter gene measurements of WNT-3A stimulated beta-catenin signalling as well as BRET assays to monitor DEP and Gq-4A recruitment and BRET-based intramolecular conformational changes. Some nice cluster analysis and molecular dynamics have then been performed to group the targeted residues into three clear groups. The main aim of the study was to understand how conformational selection can determine pathway selectivity downstream of FZDs. The main conclusion is that FZDs prefer coupling to DVL as a transducer.

The one concern I have is that the authors have predominantly applied this analysis to constitutive receptor coupling/signalling in the absence of agonist. The only situation in which agonist-stimulation was applied was in the case of the reporter gene response that measured beta-catenin-dependent gene transcription. As a consequence, one is left with the question of whether WNT-3A activation changes the coupling to DEP and Gq-alpha and alters the conformational changes observed with the intramolecular BRET sensor? Surely this is worth adding into this manuscript to get a clear view of the role of the identified microswitches in mediating agonist-induce pathway selectivity?

We thank this reviewer for the overall positive comments and for this insightful comment. Indeed, the knowledge whether WNT stimulation affects DVL interaction or G protein coupling is interesting. However, for this information, we would like to refer the reviewer to two papers published in Science Signaling where we address WNT-induced effects on FZD₅-Gq coupling (Wright, Canizal et al. 2018) and FZD₅-DVL/DEP interaction (Bowin, Kozielowicz et al. 2023) (DOI: 10.1126/scisignal.abo4974). These two papers argue that receptor-transducer interaction is indeed dynamic in response to agonist (WNT) stimulation even though G protein and DVL follow different patterns of interaction and dynamics. G proteins – in agreement with what is known for GPCRs – dissociate upon activation, whereas overexpressed DVL forms a complex with FZD₅, which undergoes a conformational rearrangement upon WNT stimulation (Bowin, Kozielowicz et al. 2023). It should be underlined that the concept of constitutive activity of GPCR is based on dynamics involving micro-switches that are essential to transmit the same information flow in response to agonist binding (Smit, Vischer et al. 2007, Zhou, Yang et al. 2019).

We have also published data on FZD₅ conformational sensors and their dynamics with WNT stimulation (Schihada, Kowalski-Jahn et al. 2021). Additionally, the FZD₅-Halo-Nluc conformational sensor responds to WNT stimulation, as shown in **Supplementary Fig. S4b**

While this is suitable for the wild-type receptor, the analysis of microswitch mutants of FZD₅ in response to agonist is less feasible. While it is possible to perform these experiments, it is so far impossible to interpret the data providing mechanistic insights: In fact, we have performed assays with the DEP biosensor and FZD₅-Halo-Nluc conformational sensors for some mutants in an early phase of the project (data shown below, normalized to basal response of each individual receptor construct):

The holistic approach to assess many/most microswitches that affect constitutive coupling in the presented readouts is hardly combinable with agonists, at least in the assays where mutations affect constitutive effector coupling. Adding agonist/WNT stimulation in an assay paradigm that is based on different baselines of the receptor mutants due to effects on transducer coupling renders it impossible to distinguish between the effect of the mutation on effector coupling vs agonist-induced changes on effector coupling. However, in the case of TOPFlash readout, baseline/constitutive activity is not measurable and thus – in connection with the extreme amplification of this assay – agonist stimulation was the only way to obtain a signal. In our eyes, it would have been indeed best to employ only assays reporting constitutive activity and the TOPFlash data indeed stick out. However, given the wide acceptance and distribution of this assay in the WNT field, we included it in this analysis.

It would also be worth knowing the actual level of expression of FZD (in the different assays; compared to endogenous expression levels) since this can obviously induce constitutive coupling at high receptor expression levels. I accept that the authors have done some good control experiments to ensure that relative cell surface expression has been monitored for each alanine mutant in each experimental set-up.

Regarding expression levels of FZD₅ in our experiments, we estimate expression levels to be manyfold higher than endogenous levels. This estimate is based on our recently published data comparing HiBiT-FZD₇ expressed from the endogenous promoter in SW480 cells as well as overexpressed from CMV promoters (Gratz, Sajkowska-Kozielewicz et al. 2023). Our experiments here are performed in HEK293A cells and FZD₁₋₁₀ KO cells provided by Benoit Vanhollebeke and we transfect FZD₅ and its mutants. The aim of this work is to understand the conserved microswitches in FZDs, which requires and justifies receptor overexpression. In connection with the argumentation above (about the difficulties to interpret WNT stimulation experiments), we argue that a potential overexpression-enhanced constitutive activity in our experimental paradigms is purely beneficial for the purpose of

the work (i) increasing the assay window and (ii) allowing to focus on the effect of the individual mutations on constitutive receptor activity.

Reviewer #2 (Remarks to the Author):

In the manuscript "Pathway selectivity in Frizzleds is achieved by conserved micro-switches defining pathway-determining, active conformations" the authors present a broad and thorough study of potential key microswitch residues identified in FZD5 via comparative analysis with known structures of GPCRs. In the study, the authors perform an alanine scan targeting functionality of these residues and assess outcomes in an extensive panel of assays reflecting various aspects of receptor behavior in Dvl-, beta-catenin- and G protein-related readouts. The accumulated data included DVL shift and interaction with DEP domain, TopFlash readout, Gq 4A-based G protein coupling readout and intrinsic conformational sensor based on HaloTag insertion in the ICL3. These results were summarized and clustered yielding individual signaling groups which were proposed to delineate structural ensembles responsible for receptor coupling to distinct pathways. Molecular dynamics simulation of receptor model was chosen as a method to investigate in more details which exact structural rearrangements might guide these preferences. Overall, the work is broad, technically sound and beyond doubt brings significantly further our understanding of the structural determinants of the functioning of the Frizzleds. However, several major and minor concerns must be addressed by the authors:

Major concerns:

- The authors begin with removal from analysis the mutants which do not show statistically significant presence at the cell surface. However, the method they have chosen (ELISA) has a clear sensitivity limit evident even on the Fig.1c and can be estimated to reliably reports surface levels in the range of >10% of that of WT protein. However, from their further data it is clear that there is little, if any, correlation between surface expression and activity in any of the assays. Therefore, even 1% of the surface expression might be sufficient to obtain measurable readouts, and thus claim that these mutants are not expressed at the cell surface might be at least misleading. Possibly, exclusion of these mutants from analysis is also not justified, though, admittedly, their aberrantly low expression levels are of concern and most likely they shouldn't be included in the analysis. Nevertheless, I suggest that the authors should perform at least one, better two functional assays of different nature, e.g. TopFlash and Gq 4A, to re-evaluate signaling potential of these mutants and reconsider their conclusions on them.

First of all, the reviewer is correct in the statement that there is little correlation between FZD surface expression and activity, e.g. DEP recruitment. We provide the following data here for discussion and would ask to remove them in case this rebuttal letter will ever be published online because the data might be included in a future manuscript. The data show FZD constructs with varying cell surface expression (bar graphs) and analysis of FZD-DEP BRET titration with increasing amounts of DEP-Venus (BRET titration, right side). Even though two of the constructs are barely expressed at the cell surface, the DEP titration curves are not distinguishable from the one obtained with the wild-type receptor, arguing that FZD-DEP BRET is not affected with largely varying cell surface expression.

Redacted.

However, the reviewer addresses indeed an interesting point. We have discussed intensively during the beginning of this project how to handle these “non-significantly expressed” mutants. A mutant that destabilizes the receptor structure in a way that affects surface expression appears functionally highly relevant. However, the data interpretation balancing the mutation’s effect on signal transmission vs structural integrity/membrane embedding is challenging. Nevertheless, the reviewer is correct that low surface expression of these mutants could yield a signal and therefore, we have attempted the following functional assays:

We have assessed the FZD₅ mutants Y^{3x33}A, W^{3x50}A, F^{5x46}A, V^{5x47}A, P^{5x50}A, K^{7x41}A, which we filtered out for further analysis due to poor cell surface expression, in the TOPFlash assay, DVL electrophoretic mobility shift, DEP titrations (BRET_{max} and BRET₅₀), and the G_q 4A coupling. The data are presented below. Observe that only two of the mutants barely shifted DVL, an assay that could serve as a second negative filter for inclusion of these mutants. We would be willing to incorporate these data into the manuscript, if indeed required. However, we have difficulties to justify inclusion given the first filtration step by reduced surface expression presented in Fig. 1 as well as the inability to assess the presented DVL shift for these mutants in a reproducible manner. These mutants show inconsistencies to generate interpretable data presumably due to reduced surface expression, especially with regard to obvious differences for differentially tagged constructs of the same receptor. As a consequence, we cannot use these data in the cluster analysis as it is only reliable and unbiased when feeding it with complete data sets. Therefore, we prefer to present the data of the selected mutants here in the rebuttal rather than incorporating them in the manuscript.

Figure information: note: the red filled bars indicate FZD₅-Nluc mutants, which were not significantly expressed at the cell surface, making the obtained values hard or even impossible to interpret.

- It is unclear how the TopFlash data, which is obtained upon stimulation by Wnt3a is combinable with the rest of the approaches which rely on ligand-free activity of the receptors. It would be logical to make a separate data set with tests of Wnt3a activity on the responsive Halo-tagged construct and of Wnt5a on Gq 4A which was shown to function in the authors' previous works. Such dataset, together with TopFlash data can be treated separately to analyze distinct roles of these microswitches in mutant-intrinsic and Wnt-responsive activities.

Please see also our elaborate response to reviewer #1 on this important point.

While we conceptually agree with this reviewer, we have – early on in this project – attempted WNT-induction e.g. for FZD-DEP interaction and FZD conformational sensors (see figures in the response to reviewer #1) and realized that the data are very difficult to interpret.

The ideal situation, in our eyes, would have been to employ constitutive readouts throughout the project. However, the TOPFlash assay is not useful in the absence of agonist stimulation compared to the other assays, because no basal response can be detected. We feel that the constitutive assessment

of signaling more directly addresses receptor function and that mutants affecting e.g. ligand binding do not affect our conclusions.

The G_q 4A read out is particular in its intrinsic GPCR coupling characteristics. The recent work by Nevin Lambert shows clearly that agonists (mostly) do not evoke a change in the BRET response when using receptors with their cognate G 4A proteins (e.g. G_q 4A for FZD₅) and thus, this assay paradigm is designed for assessment of constitutively active GPCRs rather than detecting complex dynamics in response to agonist (Jang, Lu et al. 2023).

- The authors' claim that they uncovered an array of microswitches universally applicable across the entire FZD family is based solely on 80% conservation cutoff. To maintain such claim they should introduce some representative mutations, e.g. the ones used in MD simulations, in at least one or better several other FZDs and demonstrate similar behavior. Admittedly, to a degree, this has been already done for residues R/K6.32 and W7.55 on FZD6 in the other work by the authors, but it clearly falls short of universality in context of new findings described in this work. Otherwise the "Frizzleds" in the title must be replaced with "Frizzled-5" and corresponding statements across the manuscript should be softened.

We agree with this reviewer, and we have performed complementary experiments with the "cornerstone mutants" for FZD₄ and FZD₁₀ – representatives of the Class F that are able to mediate WNT/β-catenin signaling (FZD_{3/6} do not do TOPflash) and couple to both DVL and heterotrimeric G proteins (G_{12/13} and G₁₃, respectively; Arthofer, Hot et al. 2016, Hot, Valnohova et al. 2017, Wright, Kozielowicz et al. 2019)– to be able to land in a broader argumentation for class-wide phenomena. **Those data are presented in the new Supplementary Figs. S15-S17 and an additional short paragraph was added to the main text.** Interestingly, the data are in agreement with the general concept and serve therefore as a substantial support for our conclusions. Thanks to the reviewer for this valuable suggestion.

- The part describing molecular dynamics analysis lacks coherence. It is stated that 2 representatives were taken from each cluster, but then all the details of resulting changes are discussed either individually or independently of clustering and no global conclusion is drawn. The data on the Supplementary Fig. 7 is underdescribed as there are clearly several interactions, which appear to be cluster-specific.

We thank the reviewer for this comment. During revision, large portions of this section were rewritten and hopefully the reviewer agrees that the section is more coherent now. Importantly, we tried to avoid the direct correlation between observations from the MDs and experimental data. We believe that although the simulations give us valuable insights on the changed dynamics of the different mutants, the setup is too limited for direct correlations and interpretations, since e.g. the influence of interacting transducer proteins is not considered. We tried to formulate this more clearly in the revised version of the text.

Regarding the interaction analysis we decided to not interpret this extensive set of data since it is very complex, difficult to correlate to experimental data and we could not find obvious cluster-specific changes in interaction patterns. However, if the reviewer found such cluster-specific interaction patterns that we missed during our analysis we would be very grateful for the correction of our statement.

- It is stated that for MD analysis 3 independent replicas were made for each mutant, but out of all the figures presented by authors this is only shown in Supplementary Fig.S6. The RMSD is characterized by a strong "saturating" effect because it measures differences from the starting structure using quadratic distances. This means that it rapidly becomes high and almost flat, not because the protein no longer changes, but because the protein continues to change populating conformations that are different from each other but are equally "distant" from the initial structure. Therefore, we cannot say that all 3 repeats are very similar using RMSD plots, but authors do not provide any other proper assessment of how reproducible were the results from independent replicas. The data shown on the Fig. 8 and Supplementary Figure S7 should be reassessed in this respect: instead of showing the averaged or pooled data, the data from independent replicas should be treated essentially as independent repeats and the difference assessed statistically. A relevant minor issue here - MD simulation results should be clearly annotated with how the replicate data was taken into account, at least in the figure legends and materials and methods.

We thank the reviewer for this very relevant objection. Although we have evaluated the per-replica contributions for the discussed distances and angles before, we agree that this data should also appear in the manuscript. We have now added per-replica plots for the discussed measurements and relevant mutants (compared to wild-type) in the **new Supplementary Figures S9-S11**. No statistical analysis was conducted, since in all cases the distribution of values and not the mean value was evaluated.

For the fingerprint figure (former Supplementary Figure S7, now S14) we decided to keep it as it was since the figure is already very difficult to evaluate and adding a statistical analysis to it would make it disproportionately more difficult. However, we added the interaction frequency data for the overall as well as per-replica analysis as Supplementary Data (.zip-folder) to allow an independent judgement of it. As already mentioned in the previous comment, we also refrained from interpreting the data in detail but kept it rather as a supplement to support the idea of conformational and dynamic differences between the different mutant receptors.

Following the reviewers request, we also stated more clearly in the methods section as well as figure legends how the data were treated during the analysis. For example, the following sentence was added to the methods section: *The trajectories of all replica of the same system were concatenated for all analyses, unless it is specifically stated that the analysis was conducted per replica.*

- The authors provide a detailed discussion of the structural effects of the mutations, with major rearrangements occurring in TM5 and TM6, which appears similar to class A and B GPCR activation and activated state of Smo; however, there seem to be no specific effects of TM3. It would be useful if authors add some discussion of similarities and differences between mechanisms of FZD5 and GPCRs of other classes.

We thank the reviewer for this comment and appreciate that it would be indeed interesting to investigate the similarities and differences of the activation mechanisms between FZD₅ and GPCRs of other classes. However, we think it is not within the scope of this manuscript, and would require more extensive studies since (1) it is still unclear how a DVL-bound receptor conformation looks like and (2) an analysis of the activation mechanism would require active receptor conformations, which were not used to initiate the MD simulations and likely not sampled during the MD simulation.

To reflect this, we decided to mention any potential correlations between conformational changes and experimental results more carefully in the revised MD results section and point out the limitations of the MD simulations more clearly.

Minor concerns:

- numeration in the text is not uniform. The authors sometime use "." and sometimes "x" as a separator for their system of residue numeration, and sometimes use simply their number in the sequence. The supplementary Figure S7 has conversion between two systems in the legend which is inconvenient, and only for the mutants, not for interaction pairs

Thanks for pointing this out. We have corrected this.

- colors on the figure 8 are given as text, which makes the figure difficult to read. The authors should put a small legend with the color coding of volumetric representations. Color coding of the clusters should be shown for residues on Fig. 8b, d, g.

Thanks for this suggestion. We have now introduced a color-coding legend as well as some additional color-coding in plots and figures, which in our eyes significantly improved the readability of the figure.

- the authors vaguely state that all selected residues are at least 80% conserved across the FZD family. It would be useful to make a supplementary figure with alignment of FZD family highlighting mutated positions

Thanks for this suggestion. **We have added a class wide alignment where we highlight the mutants used in this study (see the new Supplementary Figure S1).**

Reviewer #3 (Remarks to the Author):

Grätz and Kowalski-Jahn et al. investigate conserved state-stabilizing residues in the FZD family of receptors and report their role in pathway selectivity. The FZD5 has served as a representative receptor to experimentally assess a panel of functional signaling readouts including FZD-mediated WNT/ β -catenin signaling, FZD-DVL interaction, FZD-G protein coupling, and conformational dynamics. The study is complemented with molecular dynamics simulations to study the molecular impact of validated mutations on receptor structure/dynamics at atomistic resolution. Based on this data, the authors test also the hypothesis of whether distinct receptor conformations determine downstream signaling for FZD5 as a representative of the FZD family. Ultimately, a clustering approach reveals unexpected correlations between different functional outcomes.

Major comments:

1. In a first step, authors predict state-stabilizing residues for mutagenesis of FZD5. Can the authors discuss and comment on the quality of this prediction related to the hit rate of identified residues with a functional impact? Moreover, does the experimentally observed functional impact correlate to the number of altered contacts when comparing different receptor states? Of specific interest are also residues that have been predicted to be state stabilizing but did not show any impact upon mutation. What is the percentage of those residues? All in all, I would encourage the authors to dedicate a whole section to this relevant topic.

Thanks a lot for this constructive comment. **We have added a new paragraph in the results section along with a new Supplementary Fig. S6 and a Supplemental Spreadsheet.** This shows that the mutations in expressing receptor mutants have an effect in on average four of the seven functional endpoints evaluated in our study (Suppl Fig. S6). Strikingly, the two mutations T^{1x53}A and R^{6x32}A affect all seven studied functional endpoints and only one mutation (W^{3x43}A) affects a single parameter (receptor conformation). The most frequent type of functional impact was observed on the receptor conformation (21 out of 26 (81%) of expressing receptors, as measured using FZD₅-Halo-Nluc sensor). The least functional impact still affected twelve (46%) of the expressing mutants and was observed on the DVL shift.

We did not observe a quantitative correlation between the experimentally observed functional impact (number of effects of mutants) and the frequency difference of contacts across inactive-/active-state receptors structures (the underlying metric of the mutation design tools). This could be because the strength of stabilization is also affected by the type of molecular interaction. This type of validation is therefore better conducted by comparing it to mutations predicted to not have an effect. Such a (successful) validation was performed in the original publication of the mutation design tools (Fig. 6 in (Hauser, Kooistra et al. 2021) and EDF 1 in (Kooistra, Munk et al. 2021).

2. Section “Conformational FZD5 sensors reveal differences in absence of agonist”. I acknowledge the effort of the authors to investigate also conformational changes. However, at the current state, this section is not very conclusive. Unfortunately, the experimental setup allows only detecting if a conformational change occurs but without further specifications. Considering the differential impact on the functional outcome upon mutation, it would have been interesting to learn what type of helix movements correspond to a distinct functional outcome.

For instance, the authors write “Interestingly, while BRETO values, most of them were significantly higher (Fig. 6b), consistent with a conformational change towards an active state.” How does this go along with the observation that numerous mutants with “conformational changes towards an active state” show reduced Gq 4A coupling in Figure 5a.

The conformational BRET sensors are indeed suitable for detection of conformational changes but do not allow extraction of what conformational changes appear. Since the BRET acceptor (HaloTag) is placed in ICL3, it is highly likely that a swing out of TM6 lays the foundation for the observed BRET changes. Nevertheless, other dynamic processes including e.g. TM3 could be involved.

Regarding the second part of the question, the reviewer is indeed pointing to a highly relevant aspect. We thank the reviewer for this comment. Indeed, the idea is that FZD-DVL and FZD-G protein complexes are based on distinct active conformations. However, our data suggest that the HALO conformational sensor of FZD₅ is not capable of distinguishing between these distinct conformations. We write in the manuscript: *“In summary, this suggests that the FZD₅ conformational sensors are not capable of distinguishing between coupling to DEP vs heterotrimeric G protein but are indeed reporters of the receptor’s ability to efficiently accommodate either DVL or G proteins. Future work is required to resolve the structural details of distinct FZD conformations that allow specific interaction with either transducer protein.”*

This is most obvious in the mutant Y^{6x53}, which shows the largest ΔBRET_0 in the set. For further dissection of conformational rearrangements, we will have to await more high resolution structural biological approaches with suitable FZD-transducer complexes.

3. Section “MD analysis supports pathway-selective conformational dynamics of FZDs”. A better representation of the data would improve the understanding of the structural data. For instance, for the G5x65A mutant, authors show a TM5 movement in Figure 8a which is quantified as distance I2x44-G5x65 in Figure 8d. This distance distribution is informative, similar to the other distribution plots in Figure 8. However, relevant information is lost by showing only the occupancy maps of TM5 and TM6. I propose depicting instead a representative structure for the average TM5-TM6 distance in cartoon superimposing the WT to the mutant. Such a depiction also allows showing (i) the side chains of the mutant and WT, (ii) the monitored TM5-TM6 distance as well as (iii) other correlated residues such as W3x50 in the same image. Such representation would also improve the understanding of the TM5-TM6 distance in Figure 8f and the related re-orientation of ICL3 as well as the TM6-TM7 positioning in Figure 8h.

We thank the reviewer for their input on a clearer display of the data in **Figure 8** and **revised the figure based on the suggestions (and remade and added several of the Supplementary Figures such as Fig. S10-S13)**. The volumetric plot over the entire structure of the receptor for wild-type and mutants was replaced by an overlay of the cartoon representation of the representative structure of the main cluster after clustering based on the TM-C α atoms over all three replicas. In this depiction we now also indicate the distances, angles and residues discussed in Figure 8, as was suggested by the reviewer. However, for the closer evaluation of certain regions of the receptor for the selected mutants we decided to keep the volumetric plots as, in our eyes, they display more clearly the variability of the protein backbone in this region. We also believe, that showing side chains for these regions would not add valuable information and rather make the depiction more complicated, especially considering that the clustering was based on the backbone and hence the side chain orientation in the depicted structure would not necessarily be representative. In any case, we think the suggestions and changes helped to make the figure clearer and easier to grasp and hope the reviewer shares this opinion with us.

Minor comments:

4. In the introduction a different numbering scheme (e.g., W7.55) is used compared to the results section (e.g., W7x55)

Thanks for pointing this out. This has been unified.

5. Please add in Figure S7 the residue numbering scheme to facilitate the reading.

Thanks for this suggestion. We adapted the residue numbering to the scheme used in the text for all residues that have an associated number according to this scheme. All other residues are marked with their number in the sequence and their location in the receptor structure. (now **Supplementary Figure S14**)

6. In the structure comparison tool of GPCRdb, when comparing “Two sets of structures”, the structure selection remains “None selected” despite having selected structures. Please correct this error.

Thanks for this comment. This has been forwarded to GPCRdb.

References

- Arthofer, E., B. Hot, J. Petersen, K. Strakova, S. Jager, M. Grundmann, E. Kostenis, J. S. Gutkind and G. Schulte (2016). "WNT Stimulation Dissociates a Frizzled 4 Inactive-State Complex with Galpha12/13." *Mol Pharmacol* **90**(4): 447-459.
- Bowin, C. F., P. Kozielowicz, L. Gratz, M. Kowalski-Jahn, H. Schihada and G. Schulte (2023). "WNT stimulation induces dynamic conformational changes in the Frizzled-Dishevelled interaction." *Sci Signal* **16**(779): eabo4974.
- Gratz, L., J. J. Sajkowska-Kozielowicz, J. Wesslowski, J. Kinsolving, L. J. Bridge, K. Petzold, G. Davidson, G. Schulte and P. Kozielowicz (2023). "NanoBiT- and NanoBiT/BRET-based assays allow the analysis of binding kinetics of WNT-3A to endogenous Frizzled 7 in a colorectal cancer model." *Br J Pharmacol*.
- Hauser, A. S., A. J. Kooistra, C. Munk, F. M. Heydenreich, D. B. Veprintsev, M. Bouvier, M. M. Babu and D. E. Gloriam (2021). "GPCR activation mechanisms across classes and macro/microscales." *Nat Struct Mol Biol* **28**(11): 879-888.
- Hot, B., J. Valnohova, E. Arthofer, K. Simon, J. Shin, M. Uhlen, E. Kostenis, J. Mulder and G. Schulte (2017). "FZD10-Galpha13 signalling axis points to a role of FZD10 in CNS angiogenesis." *Cell Signal* **32**: 93-103.
- Jang, W., S. Lu, X. Xu, G. Wu and N. A. Lambert (2023). "The role of G protein conformation in receptor-G protein selectivity." *Nat Chem Biol*.
- Kooistra, A. J., C. Munk, A. S. Hauser and D. E. Gloriam (2021). "An online GPCR structure analysis platform." *Nat Struct Mol Biol* **28**(11): 875-878.
- Schihada, H., M. Kowalski-Jahn, A. Turku and G. Schulte (2021). "Deconvolution of WNT-induced Frizzled conformational dynamics with fluorescent biosensors." *Biosens Bioelectron* **177**: 112948.
- Smit, M. J., H. F. Vischer, R. A. Bakker, A. Jongejan, H. Timmerman, L. Pardo and R. Leurs (2007). "Pharmacogenomic and structural analysis of constitutive g protein-coupled receptor activity." *Annu Rev Pharmacol Toxicol* **47**: 53-87.
- Wright, S. C., M. C. A. Canizal, T. Benkel, K. Simon, C. Le Gouill, P. Matricon, Y. Namkung, V. Lukasheva, G. M. Konig, S. A. Laporte, J. Carlsson, E. Kostenis, M. Bouvier, G. Schulte and C. Hoffmann (2018). "FZD5 is a Galphaq-coupled receptor that exhibits the functional hallmarks of prototypical GPCRs." *Sci Signal* **11**(559).
- Wright, S. C., P. Kozielowicz, M. Kowalski-Jahn, J. Petersen, C. F. Bowin, G. Slodkowicz, M. Marti-Solano, D. Rodriguez, B. Hot, N. Okashah, K. Strakova, J. Valnohova, M. M. Babu, N. A. Lambert, J. Carlsson and G. Schulte (2019). "A conserved molecular switch in Class F receptors regulates receptor activation and pathway selection." *Nat Commun* **10**(1): 667.
- Zhou, Q., D. Yang, M. Wu, Y. Guo, W. Guo, L. Zhong, X. Cai, A. Dai, W. Jang, E. I. Shakhnovich, Z. J. Liu, R. C. Stevens, N. A. Lambert, M. M. Babu, M. W. Wang and S. Zhao (2019). "Common activation mechanism of class A GPCRs." *Elife* **8**.

REVIEWERS' COMMENTS

Reviewer #1 (Remarks to the Author):

The authors have adequately replied to my previous comments.

Reviewer #2 (Remarks to the Author):

The authors have addressed the raised issues in their entirety in the revised manuscript. Overall, the manuscript is now substantially more coherent and complete and can surely be recommended for publication. However, the authors should consider the following suggestions for introduction in the final version of the manuscript:

- the arguments and additional data on mutants with low surface expression provided by the authors are very well taken. However, I recommend softening the statement in line 121 from strict "were not expressed" to instead say that their levels merely fell below the detection limit of the method. As four of these mutants are actually active in the TopFlash assay, I also encourage authors to include at least the TopFlash data (panel c in their rebuttal figure) on these mutants as a supplement to the manuscript to avoid misleading the readers that their low surface expression means a lack of activity.
- by my comment regarding Supplementary Figure S14 I merely wished to ensure that authors fully explored this dataset, statistically or based on cut-off, for interactions that might have common patterns relative to clusters. I am obviously not in a position to conduct an in-depth analysis of these data, but from my evaluation of this figure, it appears that 209:L-308:EL1 interaction is of higher frequency in the blue cluster, while 8x49-8x52 is lower in the purple one. Even if the authors confirm that these (or perhaps others that I've missed) commonalities don't clearly relate to the selectivity or are difficult to explain, I think it would be important to include in the figure legend a statement that such analysis was performed.

Reviewer #3 (Remarks to the Author):

After carefully considering the revisions made by the authors, I can confirm that all my concerns have been adequately addressed. The authors' efforts in addressing each point have resulted in a substantially improved manuscript which makes now a significant contribution to the field.

Rebuttal letter

In summary, we appreciate the constructive and qualified criticism raised by all three reviewers. This peer review process has indeed sharpened and improved the manuscript.

REVIEWERS' COMMENTS

Reviewer #1 (Remarks to the Author):

The authors have adequately replied to my previous comments.

We are indeed happy that the revised manuscript addressed all the comments of this reviewer.

Reviewer #2 (Remarks to the Author):

The authors have addressed the raised issues in their entirety in the revised manuscript. Overall, the manuscript is now substantially more coherent and complete and can surely be recommended for publication. However, the authors should consider the following suggestions for introduction in the final version of the manuscript:

- the arguments and additional data on mutants with low surface expression provided by the authors are very well taken. However, I recommend softening the statement in line 121 from strict "were not expressed" to instead say that their levels merely fell below the detection limit of the method. As four of these mutants are actually active in the TopFlash assay, I also encourage authors to include at least the TopFlash data (panel c in their rebuttal figure) on these mutants as a supplement to the manuscript to avoid misleading the readers that their low surface expression means a lack of activity.
- by my comment regarding Supplementary Figure S14 I merely wished to ensure that authors fully explored this dataset, statistically or based on cut-off, for interactions that might have common patterns relative to clusters. I am obviously not in a position to conduct an in-depth analysis of these data, but from my evaluation of this figure, it appears that 209:L-308:EL1 interaction is of higher frequency in the blue cluster, while 8x49-8x52 is lower in the purple one. Even if the authors confirm that these (or perhaps others that I've missed) commonalities don't clearly relate to the selectivity or are difficult to explain, I think it would be important to include in the figure legend a statement that such analysis was performed.

Thanks a lot for the constructive criticism throughout the review process. We also feel that the manuscript improved substantially. We have added the TOPFlash data to the supplement as Supplementary Fig. S2 and we also included a comment as suggested in the new Fig. S15.

Reviewer #3 (Remarks to the Author):

After carefully considering the revisions made by the authors, I can confirm that all my concerns have been adequately addressed. The authors' efforts in addressing each point have resulted in a substantially improved manuscript which makes now a significant contribution to the field.

Thanks for the kind words and we indeed hope that the field will embrace the concept pushed forward in our work.